



**Sensitivity of a leaf gas-exchange model for estimating paleoatmospheric CO$_2$**
**concentration**
**Dana L. Royer[1], Kylen M. Moynihan[1], Melissa L. McKee[1], Liliana Londoño[2], and Peter J. Franks[3]**
[1]Department of Earth and Environmental Sciences, Wesleyan University, Middletown, Connecticut, USA
[2]Smithsonian Tropical Research Institute, Balboa, Ancón, Republic of Panamá
[3]Faculty of Agriculture and Environment, University of Sydney, Sydney, New South Wales, Australia
**Correspondence:** Dana L. Royer (droyer@wesleyan.edu)
**Abstract.** Leaf gas-exchange models show considerable promise as paleo-CO$_2$ proxies. They are largely
mechanistic in nature, provide well-constrained estimates even when CO$_2$ is high, and can be applied to
most subaerial, stomata-bearing leaves from C$_3$ taxa, regardless of age or taxonomy. Here we place
additional observational and theoretical constraints on one of these models, the "Franks" model. In
order to gauge the model's general accuracy in a way that is appropriate for fossil studies, we estimated
CO$_2$ from 40 species of extant angiosperms, conifers, and ferns based only on measurements that can be
made directly from fossils (leaf $\delta^{13}$C and stomatal density and size) and a limited sample size (1-3 leaves
per species). The mean error rate is 28%, which is similar to or better than the accuracy of other leading
paleo-CO$_2$ proxies. We find that leaf temperature and photorespiration do not strongly affect estimated
CO$_2$, although more work is warranted on the possible influence of O$_2$ concentration on
photorespiration. Leaves from the lowermost 1-2 m of closed-canopy forests should not be used
because the local air $\delta^{13}$C value is lower than the global well-mixed value. Such leaves are not common
in the fossil record, but can be identified by morphological and isotopic means.
**1 Introduction**
Leaves on terrestrial plants are well poised to record information about the concentration of
atmospheric CO$_2$. They are in direct contact with the atmosphere and have large surface-area-to-volume
ratios, so the leaf internal CO$_2$ concentration is tightly coupled to atmospheric CO$_2$ concentration. Also,
leaves are specifically built for the purpose of fixing atmospheric carbon into structural tissue, and face
constant selection pressure to optimize their carbon uptake relative to water loss. As a result, many
components of the leaf system are sensitive to atmospheric CO$_2$, and these components feedback on
one another to reach a new equilibrium when atmospheric CO$_2$ changes. In terms of carbon assimilation,
Farquhar and Sharkey (1982) modeled this system in its simplest form as:
$$A_n = g_{c(tot)} \times (c_a - c_i),\qquad\qquad(1)$$
where $A_n$ is the leaf CO$_2$ assimilation rate (µmol m$^{-2}$ s$^{-1}$), $g_{c(tot)}$ is the total operational conductance to CO$_2$
diffusion from the atmosphere to site of photosynthesis (mol m$^{-2}$ s$^{-1}$), $c_a$ is atmospheric CO$_2$
concentration (µmol mol$^{-1}$ or ppm), and $c_i$ is leaf intercellular CO$_2$ concentration (µmol mol$^{-1}$ or ppm)
(see also Von Caemmerer, 2000).
Rearranging Eq. (1) for atmospheric CO$_2$ yields:
$$c_a = \frac{A_n}{g_{c(tot)} \times (1 - \frac{c_i}{c_a})}.\qquad\qquad(2)$$



Equation (2) forms the basis of two leaf gas-exchange approaches for estimating paleo-$CO_2$ from fossils
(Konrad et al., 2008; Franks et al., 2014; Konrad et al., 2017). In the Franks model, conductance is
estimated in part from measurements of stomatal size and density, $c_i/c_a$ from measurements of leaf $\delta^{13}C$
along with reconstructions of coeval air $\delta^{13}C$ (see also Eq. 9), and $A_n$ from knowledge of living relatives
and its dependency on $c_a$ (Franks et al., 2014). Following Farquhar et al. (1980), the latter is modeled as
(Franks et al., 2014; Kowalczyk et al., 2018):

$$A_n = A_0 \frac{[(\frac{c_i}{c_a})c_a - \Gamma^*][(\frac{c_{i0}}{c_{a0}})c_{a0} + 2\Gamma^*]}{[(\frac{c_i}{c_a})c_a + 2\Gamma^*][(\frac{c_{i0}}{c_{a0}})c_{a0} - \Gamma^*]},$$
(3)


where $\Gamma^*$ is the $CO_2$ compensation point in the absence of dark respiration (ppm) and the subscript "0"
refers to conditions at a known $CO_2$ concentration (typically present-day). Equations (2) and (3) are then
solved iteratively until the solution for $c_a$ converges.
These gas-exchange approaches grew out of a group of paleo-$CO_2$ proxies based on the $CO_2$
sensitivity of stomatal density ($D$) or the similar metric stomatal index (Woodward, 1987; Royer, 2001).
Here, the $D$-$c_a$ sensitivity is calibrated in an extant species, allowing paleo-$CO_2$ inference from the same
(or very similar) fossil species. These empirical relationships typically follow a power-law function
(Wynn, 2003; Franks et al., 2014; Konrad et al., 2017):

$$c_a = \frac{1}{kD^\alpha},$$
(4)


where $k$ and $\alpha$ are species-specific constants.
The related stomatal ratio proxy is simplified: $D$ is measured in an extant species ($D_0$, at present-
day $c_{a0}$) and then the ratio of $D_0$ to $D$ in a related fossil species is assumed to be linearly related to the
ratio of paleo-$c_a$ to present-day $c_{a0}$ (Chaloner and McElwain, 1997; McElwain, 1998):

$$\frac{c_a}{c_{a0}} = k \frac{D_0}{D}.$$
(5)


Equation (5) can be rearranged to match Eq. (4) but with $\alpha$ fixed at 1. Thus, paleo-$CO_2$ estimates using
the stomatal ratio proxy are based on a one-point calibration and an assumption that $\alpha$ = 1;
observations do not always support this assumption (e.g., $\alpha$ = 0.43 for *Ginkgo biloba*; Barclay and Wing,
2016). The scalar $k$ was originally set at 2 for Paleozoic and Mesozoic reconstructions so that paleo-$CO_2$
estimates during the Carboniferous matched that from long-term carbon cycle models (Chaloner and
McElwain, 1997). For younger reconstructions, $k$ is probably closer to 1 (by definition, $k$ = 1 for present-
day plants). We note that the stomatal ratio proxy was originally conceived as providing qualitative
information, only, about paleo-$CO_2$ (McElwain and Chaloner, 1995, 1996; Chaloner and McElwain, 1997;
McElwain, 1998) and has not been tested with dated herbaria materials or with $CO_2$ manipulation
experiments.
At high $CO_2$, the $D$-$c_a$ sensitivity saturates, leading to uncertain paleo-$CO_2$ estimates, often with
unbounded upper limits (e.g., Smith et al., 2010; Doria et al., 2011). Stomatal density does not respond
to $CO_2$ in all species (Woodward and Kelly, 1995; Royer, 2001), and because $D$-$c_a$ relationships can be
species-specific (that is, different species in the same genus with different responses; Beerling, 2005;
Haworth et al., 2010), only fossil taxa that are still alive today should be used. The gas-exchange proxies
partly address these limitations: 1) $CO_2$ estimates remain well-bounded—even at high $CO_2$—and their
precision is similar to or better than other leading paleo-$CO_2$ proxies (~+35/-25% at 95% confidence;
Franks et al., 2014); 2) the models are mostly mechanistic; that is, they are explicitly driven by plant





physiological principles, not just empirical relationships measured on living plants; 3) because the
models retain sensitivity at high $CO_2$ and do not require that a fossil species still be alive today, much of
the paleobotanical record is open for $CO_2$ inference, regardless of age or taxonomy; and 4) because the
models are based on multiple inputs linked by feedbacks, they can still perform adequately even if one
or more of the inputs in a particular taxon is not sensitive to $CO_2$, for example stomatal density (Milligan
et al., in review).
We note that the published uncertainties (= precision) associated with the stomatal density
proxies are probably too small because they usually only reflect uncertainty in the calibration regression
or in the measured values of fossil stomatal density, but not both; when this is done, errors often
exceed ±30% at 95% confidence (Beerling et al., 2009). Also, error rates in estimates from extant taxa
where $CO_2$ is known (= accuracy) are usually smaller with the stomatal density proxies (e.g., Barclay and
Wing, 2016), but this is expected because the same taxa have been calibrated in present-day (or near
present-day) conditions. Because the gas-exchange proxies are largely built from physiological
principles, they have less "recency" bias; that is, the gas-exchange proxies estimate present-day and
paleo-$CO_2$ with similar certainty when the same methods are used to determine the inputs.
**2 Study Aims and Methods**
Leaf gas-exchange proxies for paleo-$CO_2$ are becoming popular (Konrad et al., 2008; Grein et al.,
2011b; Grein et al., 2011a; Erdei et al., 2012; Roth-Nebelsick et al., 2012; Grein et al., 2013; Franks et al.,
2014; Maxbauer et al., 2014; Roth-Nebelsick et al., 2014; Montañez et al., 2016; Reichgelt et al., 2016;
Konrad et al., 2017; Tesfamichael et al., 2017; Kowalczyk et al., 2018; Lei et al., 2018; Londoño et al.,
2018; Richey et al., 2018; Milligan et al., in review). However, many elements of these models remain
understudied. Here we investigate four such elements for the Franks et al. (2014) model: how does the
model perform across a large number of phylogenetically diverse taxa; and how is the model affected by
temperature, photorespiration, and proximity to the forest floor? We describe next the motivation and
details of the study design.
2.1 General testing in living plants
Franks et al. (2014) tested the model on four species of field-grown trees (three gymnosperms and one
angiosperm) and one conifer grown in chambers at 480 and 1270 ppm $CO_2$. The average error rate
(absolute value of estimated $CO_2$ minus measured $CO_2$, divided by measured $CO_2$) was 5%. Follow-up
work with three field-grown tree species (Maxbauer et al., 2014; Kowalczyk et al., 2018), $CO_2$
experiments on seven tropical trees species (Londoño et al., 2018), and experiments on two fern and
one conifer species (Milligan et al., in review) indicate somewhat higher error rates (Fig. 1). Combined,
the average error rate is 19% (median = 13%).





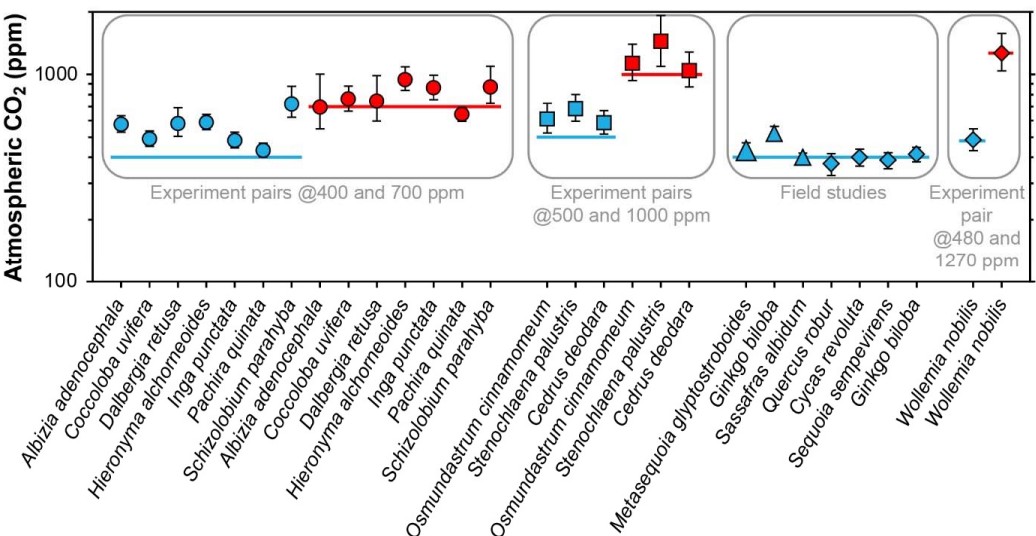

**Figure 1.** Published $CO_2$ estimates using the Franks model for extant plants where the physiological inputs $A_0$ (assimilation rate at a known $CO_2$ concentration) and/or $g_{c(op)}/g_{c(max)}$ (ratio of operational to maximum leaf conductance to $CO_2$) were measured directly. Horizontal lines are the correct $CO_2$ concentrations. Uncertainties in the estimates correspond to the 16th-84th percentile range. Circles are from Londoño et al. (2018), squares from Milligan et al. (in review), large triangle from Maxbauer et al. (2014), small triangles from Kowalczyk et al. (2018), and diamonds from Franks et al. (2014).

In these studies, two of the key physiological inputs were measured directly with an infrared gas analyzer: the assimilation rate at a known $CO_2$ concentration ($A_0$) and/or the ratio of operational to maximum stomatal conductance to $CO_2$ ($g_{c(op)}/g_{c(max)l}$, or $\zeta$), the latter of which is important for calculating the total leaf conductance ($g_{c(tot)}$). These two inputs cannot be directly measured on fossils; thus, the error rates associated with Figure 1 may not be representative for fossil studies. Franks et al. (2014) argue that within plant functional types growing in their natural environment, mean $A_0$ is fairly conservative, leading to the recommended mean $A_0$ values in Franks et al. (2014) (12 µmol m$^{-2}$ s$^{-1}$ for angiosperms, 10 for conifers, and 6 for ferns and ginkgos). Along similar lines, the mean ratio $g_{c(op)}/g_{c(max)}$ tends to be conserved across plant functional types; Franks et al. (2014) recommend a value of 0.2, which may correspond to the most efficient setpoint for stomata to control conductance (Franks et al., 2012). This conservation of physiological function is one of the underlying principles in the Franks model.

Here we test this assumption by estimating $CO_2$ from 40 phylogenetically diverse species of field-grown trees. In making these estimates, we use the recommended mean values of $A_0$ and $g_{c(op)}/g_{c(max)}$ from Franks et al. (2014) instead of measuring them directly. Thus, this dataset should be a more faithful gauge for model accuracy as applied to fossils. Of the 40 species, 21 were previously published in Londoño et al. (2018), who collected sun-adapted canopy leaves of angiosperms using a crane in Parque Nacional San Lorenzo, Panama. To test the method in temperate forests, we collected leaves from eleven angiosperm and seven conifer species from Dinosaur State Park (Rocky Hill, Connecticut), Wesleyan University (Middletown, Connecticut), and Connecticut College (New London,





Connecticut) during the summer of 2015. Here, all trees grew in open, park-like settings; one to three
sun leaves were sampled from the lower outside crown of each tree. In January of 2015, we also
sampled sun-exposed leaves from the tree fern *Cyathea arborea* in El Yunque National Forest, Puerto
Rico (near the Yokahú Tower).
Stomatal size and density were measured either on untreated leaves using epifluorescence
microscopy with a 420-490 nm filter, or on cleared leaves (using 50% household bleach or 5% NaOH)
using transmitted-light microscopy. For most species, whole-leaf $\delta^{13}$C comes from Royer and Hren
(2017); the same leaves were measured for $\delta^{13}$C and stomatal morphology. The UC Davis Stable Isotope
Facility measured some additional leaf samples. Table S1 summarizes for these 40 species all of the
inputs needed to run the Franks model, along with the estimated $CO_2$ concentrations. Uncertainties in
the estimates are based on error propagation using Monte Carlo simulations (Franks et al., 2014).
2.2 Temperature
The Franks model can be configured for any temperature. Franks et al. (2014) recommend that the
photosynthesis parameters $A_0$ and $\Gamma^*$, and the air physical properties affecting diffusion of $CO_2$ into the
leaf (the ratio of $CO_2$ diffusivity in air to the molar volume of air, or $d/v$) correspond with the mean
daytime growing-season leaf temperature (more precisely, assimilation-weighted leaf temperature). The
reasoning behind this is that (i) the assimilation-weighted leaf temperature will correspond with the
mean $c_i/c_a$ derived from fossil leaf $\delta^{13}$C; and (ii) both theory (Michaletz et al., 2015; Michaletz et al.,
2016) and observations (Helliker and Richter, 2008; Song et al., 2011) indicate that the control of leaf
gas exchange leads to relatively stable assimilation-weighted leaf temperatures (~19-25 °C from
temperate to tropical regions) despite large differences in air temperature. This is mostly due to the
effects of transpiration on leaf energy balance. Franks et al. (2014) chose a fixed temperature of 25 °C
because much of the Mesozoic and Cenozoic correspond to climates warmer than the present-day.
When applying the Franks model to known cooler paleoenvironments, improved accuracy may be
achieved with leaf-temperature-appropriate values for $A_0$, $\Gamma^*$, and $d/v$.
Bernacchi et al. (2003) proposed the following temperature sensitivity for $\Gamma^*$ based on
experiments:
$\Gamma^* = e^{\left(19.02 - \frac{37.83}{RT}\right)}$,          (6)
where $R$ is the molar gas constant (8.31446×10$^{-3}$ kJ K$^{-1}$ mol$^{-1}$) and $T$ is leaf temperature (K). Marrero and
Mason (1972) describe the sensitivity of water vapor diffusivity to temperature as:
$d = 1.87 \times 10^{-10} \left(\frac{T^{2.072}}{P}\right)$,          (7)
where $P$ is atmospheric pressure, which we fix at 1 atmosphere. Lastly, the temperature sensitivity of
the molar volume of air follows ideal gas principles:
$v = v_{STP} \left(\frac{T}{T_{STP}}\right)\left(\frac{P}{P_{STP}}\right)$,          (8)
where $T_{STP}$ is 273.15 K, $P_{STP}$ is 1 atmosphere, and $v_{STP}$ is the air volume at $T_{STP}$ and $P_{STP}$ (0.022414 m$^3$ mol$^{-1}$
).
Using Eqs. (6-8), we can describe how, conceptually, the sensitivities of $\Gamma^*$ and $d/v$ to leaf
temperature affect estimates of $CO_2$ from the Franks model. We apply these relationships to a suite of





| 409 fossil and extant leaves from 62 species of angiosperms, gymnosperms, and ferns. These data come
| from the current study (see Sect. 2.1 and 2.4) and Londoño et al. (2018), Kowalczyk et al. (2018), and
| Milligan et al. (in review).
| To experimentally test more generally how the Franks model is influenced by temperature, we
| grew six species of plants inside two growth chambers with contrasting temperatures (Conviron E7/2;
| Winnipeg, Canada). Air temperature was 28 °C and 20 °C during the day, and 19 °C and 11 °C during the
| night. We note that the difference in leaf temperature was probably smaller than that in air
| temperature during the day (8 °C; see earlier discussion). We held fixed the day length (17 hours with a
| 30 minute simulated dawn and dusk) and $CO_2$ concentration (500 ppm). Humidity differed moderately
| between chambers (76.5 ± 1.8% 1σ and 90.0 ± 3.6%). To minimize any chamber effects, we alternated
| plants between chambers every two weeks.
| Four of the species started as saplings purchased from commercial nurseries: bare-root, one-
| foot tall saplings of *Acer negundo* and *Carpinus caroliniana*, one-foot tall saplings of *Ostrya virginiana*
| with a soil ball, and bare-root, four-inch tall saplings of *Ilex opaca*. We grew the other two species from
| seed: *Betula lenta* from a commercial source, and *Quercus rubra* from a single tree on Wesleyan
| University's campus. All seeds were soaked in water for 24 hours and then cold stratified in a
| refrigerator for 30 and 60 days, respectively.
| All seeds and saplings grew in the same potting soil (Promix Bx with Mycorise; Premier
| Horticulture; Quakertown, Pennsylvania, USA) and fertilizer (Scotts all-purpose flower and vegetable
| fertilizer; Maryville, Ohio, USA). They were watered to field capacity every other day, and we discarded
| any excess water passing through the pots. After three months of growth in the chambers, for each
| species-chamber pair we harvested the three newest fully expanded leaves whose buds developed
| during the experiment. In most cases, we harvested five plants per species-chamber pair; the one
| exception was *I. opaca*, where we were limited to three plants in the warm treatment and two in the
| cool treatment.
| We measured stomatal size and density on cleared leaves (using 50% household bleach) with
| transmitted-light microscopy. Whole-leaf $\delta^{13}C$ comes from the UC Davis Stable Isotope Facility and the
| Light Stable Isotope Mass Spec Lab at the University of Florida; the same leaves were measured for $\delta^{13}C$
| and stomatal morphology. Because we used the same $CO_2$ gas cylinder as Milligan et al. (in review), we
| used their two-end-member mixing model to calculate the $\delta^{13}C$ of the chamber $CO_2$ at 500 ppm (-10.6
| ‰). We used the recommended values from Franks et al. (2014) for the physiological inputs $A_0$ and
| $g_{c(op)}/g_{c(max)}$. Table S1 summarizes all of the inputs from this experiment needed to run the Franks model,
| along with the estimated $CO_2$ concentrations. The standard errors for the inputs are based on plant
| means.
| To test if leaf $\delta^{13}C$ and stomatal morphology (stomatal density, stomatal pore length, and single
| guard cell width) differed between temperature treatments across species, we implemented a mixed
| model in R (R Core Team, 2016) using the lme4 (Bates et al., 2015) and lmerTest (Kuznetsova et al.,
| 2017) packages, with temperature and species as the two fixed factors. To test if there was a significant
| difference between $CO_2$ estimates from the two temperature treatments, we ran a Kolmogorov–
| Smirnov (KS) test in R. For each species, we first estimated $CO_2$ for each plant in the warm and cool
| treatments based on simulated inputs constrained by their means and variances. In the typical case with
| five plants per chamber, this produced five $CO_2$ estimates for the warm chamber and the same for the
| cool chamber. A KS test was then used to test for a significant temperature effect. We repeated this
| procedure 10,000 times, with 10,000 associated KS tests. The fraction of tests with a p-value < 0.05 was
| taken as the overall p value. An advantage of this approach is that it incorporates both within- and
| across-plant variation.




2.3 Photorespiration

$c_i/c_a$ is estimated in the Franks model following Farquhar et al. (1982):

$$\Delta_{leaf} = a + (b - a) \times \frac{c_i}{c_a}, \tag{9}$$

where $a$ is the carbon isotope fractionation due to diffusion of $CO_2$ in air (4.4‰; Farquhar et al., 1982), $b$ is the fractionation associated with RuBP carboxylase (30‰; Roeske and O'Leary, 1984), and $\Delta_{leaf}$ is the net fractionation between air and assimilated carbon ([$\delta^{13}C_{air}$ - $\delta^{13}C_{leaf}$]/[1+$\delta^{13}C_{leaf}$/1000]).
    Equation (9) can be expanded to include other effects, including photorespiration (Farquhar et al., 1982):

$$\Delta_{leaf} = a + (b - a) \times \frac{c_i}{c_a} - \frac{f\Gamma*}{c_a}, \tag{10}$$

where $f$ is the carbon isotope fractionation due to photorespiration. Photorespiration occurs when the enzyme rubisco fixes $O_2$, not $CO_2$ (i.e., RuBP oxygenase). One product of photorespiration is $CO_2$ (Jones, 1992), whose $\delta^{13}C$ is lower than the source substrate glycine. If this respired $CO_2$ escapes to the atmosphere, the $\delta^{13}C$ of the leaf carbon becomes more positive. Thus, if $c_i/c_a$ is calculated using Eq. (9), as is common practice, the calculation may be falsely low, leading to an underprediction of atmospheric $CO_2$.
    Measured values for $f$ vary from ~9-15‰ (see compilation in Schubert and Jahren, 2018), which is in line with theoretical predictions (Tcherkez, 2006). At a 400 ppm atmospheric $CO_2$ and $\Gamma*$ of 40 ppm, Eq. (10) implies that ~1‰ of $\Delta_{leaf}$ is due to photorespiration, meaning that $c_i/c_a$ should be ~0.04 higher relative to Eq. (9). Here, using the suite of fossil and extant leaves described in Sect. 2.2, we explore how the carbon isotopic fractionation associated with photorespiration affects $CO_2$ estimates with the Franks model. Because $c_i/c_a$ is present in both of the fundamental equations (Eqs. 2 and 3), we solve them iteratively until $c_i/c_a$ converges.

2.4 Leaves that grow close to the forest floor

The composition of air close to the forest floor can differ considerably from the well-mixed atmosphere. Of relevance to the Franks model, soil respiration can lead to a locally higher $CO_2$ concentration and lower $\delta^{13}C_{air}$ (Table 1). This effect is strongest at night, when the forest boundary layer is thickest (e.g., Munger and Hadley, 2017), but we focus here on daylight hours because that is when most plants take up $CO_2$. In wet tropical forests, which can have very high soil respiration rates, $CO_2$ during the day near the forest floor can be elevated by tens-of-ppm, and the $\delta^{13}C_{air}$ can be 2-3‰ lower; in temperate forests, the deviations are smaller (Table 1). Above ~2 m, $CO_2$ concentrations and air $\delta^{13}C$ during the daytime largely match the well-mixed atmosphere.





**Table 1.** Deviations in the $\delta^{13}C$ and concentration of $CO_2$ close to a forest floor relative to well-mixed air above the canopy. All measurements were made close to mid-day.

| Study | $\delta^{13}C_{air}$ relative to well-mixed air (‰) | $CO_2$ relative to well-mixed air (ppm) | Height above forest floor (m) | Forest location |
|---|---|---|---|---|
| **Tropical forest** | | | | |
| Broadmeadow et al. (1992) | -2 | +20 | 0.15-1 | Trinidad during dry season |
| Buchmann et al. (1997) | -2 | +30 | 0.70-0.75 | French Guiana during wet and dry seasons |
| Holtum and Winter (2001) | NA | +50 | 0.10 | Panama during wet and dry seasons |
| Lloyd et al. (1996) | -3 | +70 | 1 | Brazil (Amazon Basin) |
| Quay et al. (1989) | -3 | +20 | 2 | Brazil (Amazon Basin) |
| Sternberg et al. (1989) | -2 | +25 | 1 | Panama during wet and dry seasons |
| | | | | |
| **Temperate forest** | | | | |
| Francey et al. (1985) | -1 | +20 | 1 | Tasmania |
| Munger and Hadley (2017) | NA | +15 | 1 | Massachusetts (Harvard Forest) |

As a result, leaves that grow close to the forest floor may cause the Franks model to produce $CO_2$ estimates higher than that of the mixed atmosphere for at least two reasons. First, the concentration of $CO_2$ near the forest floor is elevated; that is, the model may correctly estimate a $CO_2$ concentration that the user is not interested in. Second, because the $\delta^{13}C_{air}$ that a forest-floor plant experiences is lower than the global well-mixed value, if the user chooses the well-mixed value for model input (inferred, for example, from the $\delta^{13}C$ of marine carbonate; Tipple et al., 2010), $c_i/c_a$ and thus atmospheric $CO_2$ will be overestimated (see Eq. 2).

We sought to test how the Franks model is affected by the forest-floor microenvironment for five tropical angiosperm species and fifteen temperate angiosperm and fern species. The tropical leaves were sampled at ~1-2 m height from Parque Nacional San Lorenzo, Panama. In contrast to the canopy data set from San Lorenzo (Sect. 2.1), these $CO_2$ estimates have not been previously reported. In the summer of 2015, seven fern species were sampled at ~0.5 m height from Connecticut College and Wesleyan University. Also, we used leaf vouchers from Royer et al. (2010), who sampled eight herbaceous angiosperm species at ~0.1-0.2 m height from Reed Gap, Connecticut. For all 20 species, stomatal and carbon isotopic measurements follow the methods described in Sect. 2.1. Table S1 contains all of the inputs needed to run the Franks model, along with the estimated $CO_2$ concentrations.

We also investigated if we could include the forest-floor $\delta^{13}C_{air}$ effect in our estimates of atmospheric $CO_2$. If the only $CO_2$ inputs close to the forest floor are from the soil and well-mixed atmosphere, the system can be modeled as a two-endmember mixing model where $\delta^{13}C_{air}$ has a positive, linear relationship with $1/CO_2$ (Keeling, 1958). If the $CO_2$ concentration and $\delta^{13}C$ of both endmembers are known, the forest-floor microenvironment should fall somewhere on the modelled line. Importantly, the Franks model provides a second constraint on the system. Here, $\delta^{13}C_{air}$ has a negative, nonlinear relationship with $1/CO_2$ because $\delta^{13}C_{air}$ is positively related to $c_i/c_a$ and $CO_2$. The Franks model thus provides a second calculation for the relationship between $\delta^{13}C_{air}$ and estimated $CO_2$





concentration. The intersection between the two curves should be the correct $\delta^{13}C_{air}$ and $CO_2$
concentration for the forest-floor microenvironment.
To estimate the soil $CO_2$ endmember, we measured the $\delta^{13}C$ of soil organic matter collected
from the A horizons of 13 soil sites at San Lorenzo, and of five each at Reed Gap and Connecticut
College. For all soils, we assume a 5000 ppm $CO_2$ concentration for a depth that is below the zone of $CO_2$
diffusion from the atmosphere (~0.3 m; Cerling, 1999; Breecker et al., 2009). The true value for wet
temperate and tropical forest soils may be somewhat less or substantially more than 5000 ppm (Medina
et al., 1986; Cerling, 1999; Hirano et al., 2003; Hashimoto et al., 2004; Sotta et al., 2004). Because the
mixing model uses $1/CO_2$, a much higher $CO_2$ concentration (e.g., 10000 ppm) has little impact on our
results.
**3 Results and Discussion**
3.1 General testing in living plants
Estimates of $CO_2$ across the 40 tree species sampled in the field range from 275 to 850 ppm, with a
mean of 478 ppm and median of 472 ppm (Fig. 2). There are no strong differences across taxonomic
orders, nor between leaves from tropical and temperate forests. The mean error rate across the
estimates is 28% (median = 24%), which is higher than estimates that include direct measurements of
the physiological inputs $A_0$ and $g_{c(op)}/g_{c(max)}$ (mean = 19%; median = 13%; Fig. 1). Along similar lines, if the
estimates presented in Fig. 1 are re-estimated using the values for $A_0$ and $g_{c(op)}/g_{c(max)}$ recommended by
Franks et al. (2014), the mean error rate increases to 31% (median = 21%).
These results indicate that $CO_2$ accuracy is generally improved when $A_0$ and/or $g_{c(op)}/g_{c(max)}$ is
measured. These measurements require expensive gas-exchange equipment and are not always easy or
practical to make. Moreover, $A_0$ and $g_{c(op)}/g_{c(max)}$ cannot be measured on fossils. Some gains in accuracy
are possible by measuring $A_0$ and $g_{c(op)}/g_{c(max)}$ on extant relatives of the fossil species (e.g., the same
genus). Absent of this, our analysis using the recommended mean values of Franks et al. (2014) indicates
an error rate, on average, of approximately 28%. This is comparable to or better than other leading
paleo-$CO_2$ proxies (Franks et al., 2014).
One reliable way to improve accuracy is to estimate $CO_2$ with multiple species because the
falsely-high and falsely-low estimates partially cancel each other out. The grand mean of estimates
presented in Fig. 2 (478 ppm) is 20% from the 400 ppm target, which is less than the 28% mean error
rate of individual estimates.





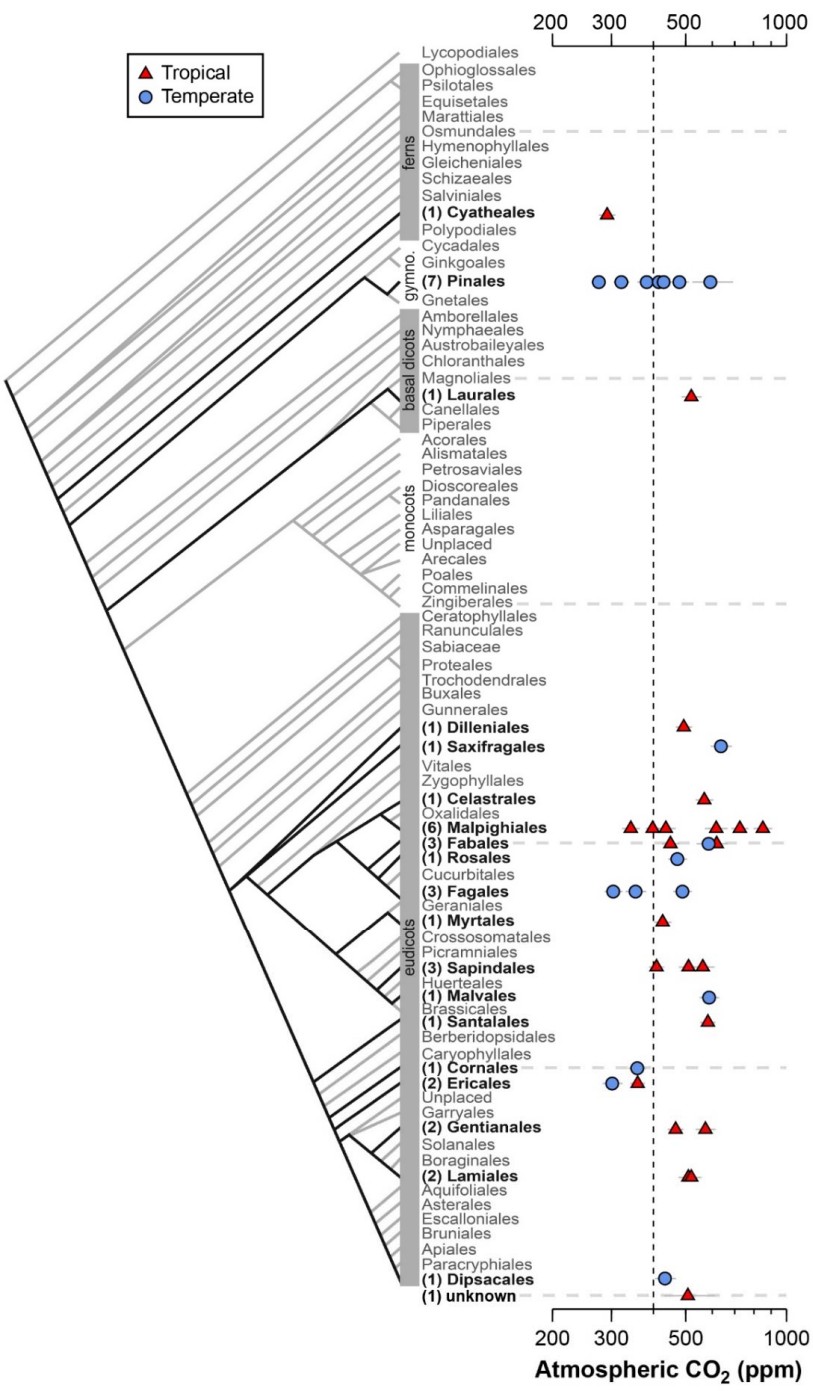

**Figure 2.** Estimates of $CO_2$ based on canopy leaves from 40 tree species. Uncertainties in the estimates correspond to the 16th-84th percentile range. Vertical line is the correct concentration (400 ppm). On the left is an order-level vascular plant phylogeny (APW v.13; Stevens, 2001 onwards).



Dow et al. (2014) observed that $g_{c(op)}/g_{c(max)}$ inversely varies with $CO_2$ in *Arabidopsis thaliana*, but
primarily at subambient concentrations (yellow triangles in Fig. 3). At elevated $CO_2$, $g_{c(op)}/g_{c(max)}$ is close
to 0.2, which is the value recommended by Franks et al. (2014). Data from eleven species of
angiosperms, conifers, and ferns at present-day (or near present-day) and elevated $CO_2$ concentrations
support the view of a limited effect at high $CO_2$ (Fig. 3; Franks et al., 2014; Londoño et al., 2018; Milligan
et al., in review). More data at subambient $CO_2$ are needed, but for $CO_2$ concentrations similar to or
higher than the present-day, we see no strong reason to include a $CO_2$ sensitivity in $g_{c(op)}/g_{c(max)}$. The
rather low values for *Cedrus deodara* and many of the tropical angiosperms (<0.1) are likely due to
stress imposed by their growth chamber environment; these $g_{c(op)}/g_{c(max)}$ values are probably not
representative of field-grown trees, which tend to be closer to 0.2 (Franks et al., 2014).

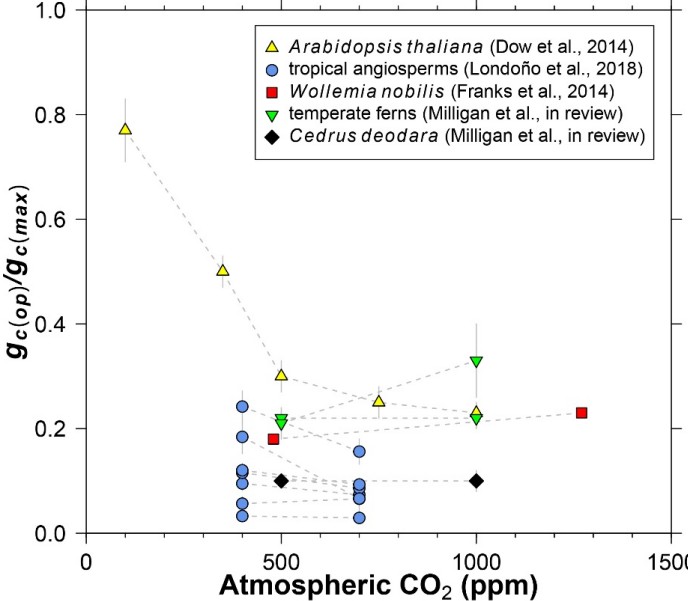

**Figure 3.** Literature compilation of the sensitivity of $g_{c(op)}/g_{c(max)}$ (ratio of operational to maximum leaf
conductance to $CO_2$) to atmospheric $CO_2$ concentration.
3.2 Temperature
The temperature sensitivities of the ratio of diffusivity of $CO_2$ in air to the molar volume of air ($d/v$) and
the $CO_2$ compensation point in the absence of dark respiration ($\Gamma^*$) have little effect on estimated $CO_2$ in
the Franks model (Fig. 4). Given that assimilation-weighted leaf temperature only varies about 7 °C
across plants today, the differences shown in Fig. 4—which are based on leaf temperatures of 25 °C and
15 °C—are likely a maximum effect. As such, we consider the use of a fixed leaf temperature (e.g., 25 °C)
in the model to be a defensible simplification.





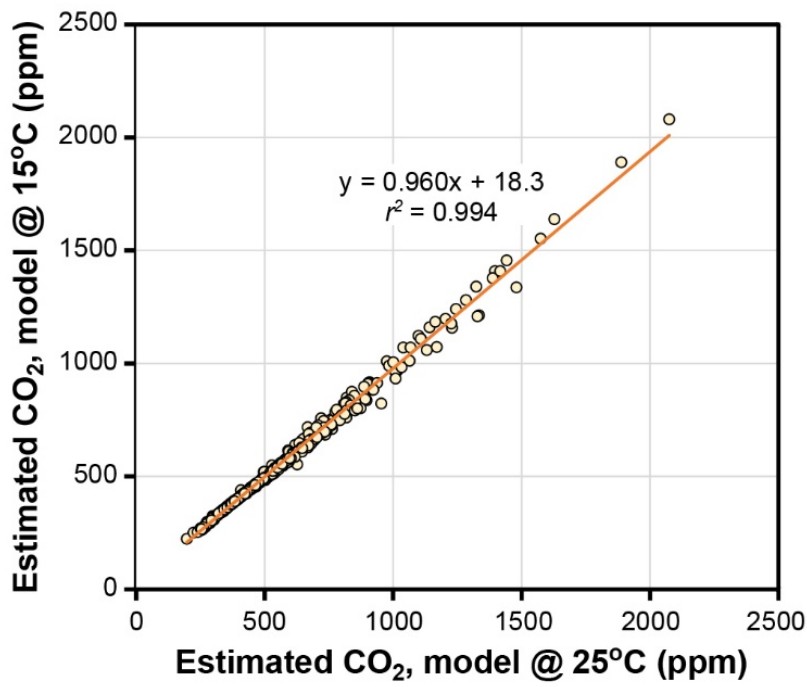

**Figure 4.** Estimates of $CO_2$ at leaf temperatures of 25 °C and 15 °C. Each symbol is an extant or fossil leaf.
The difference in estimated $CO_2$ for any leaf is due to the theoretical effects of temperature on gas
diffusion ($d/v$) and the $CO_2$ compensation point in the absence of dark respiration ($\Gamma^*$) (Eqs. 6-8).

Other inputs in the model may respond to temperature, though. In our growth chamber
experiments where daytime air temperatures were 28 °C and 20 °C, the effect on estimated $CO_2$ was
mixed (Fig. 5). In five out of six species, estimated $CO_2$ was higher in the warm treatment, but for all
species these differences were not statistically significant ($P > 0.05$ based on a KS test; see Methods).
Incorporating the temperature sensitivities in $d/v$ and $\Gamma^*$ had little effect ("adj" estimates in Fig. 5), as
expected from Fig. 4.
None of the measured inputs—stomatal density, stomatal pore length, single guard cell width,
and leaf $\delta^{13}C$—were significantly affected by temperature across all species ($P > 0.05$ for each of the four
inputs based on a mixed model; see Methods). These small differences probably cannot account for the
differences in estimated $CO_2$ between temperatures. It is more likely that some of the inputs that we did
not directly measure, such as assimilation rate ($A_0$), the $g_{c(op)}/g_{c(max)}$ ratio, or mesophyll conductance ($g_m$),
differ from the true mean value. In the cases for the five species where estimated $CO_2$ is higher in the
warm treatment, our mean $A_0$ for the warm plants must be falsely high, or $g_{c(op)}/g_{c(max)}$ or $g_m$ falsely low.
In summary, we see no strong reason to expand the parameterization of temperature in the
model, though more growth-chamber experiments may be warranted. We note that the across-species
means of estimated $CO_2$ for the warm and cool treatments are reasonably close to the 500 ppm target
(590 and 502 ppm, respectively) and overall have a mean error rate of 25%. This level of uncertainty is
similar to our field estimates where we did not measure $A_0$ or $g_{c(op)}/g_{c(max)}$ (28%; see Fig. 2). This too
provides support for our recommendation that it is not critical to include a broader treatment of
temperature in the model.

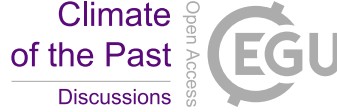

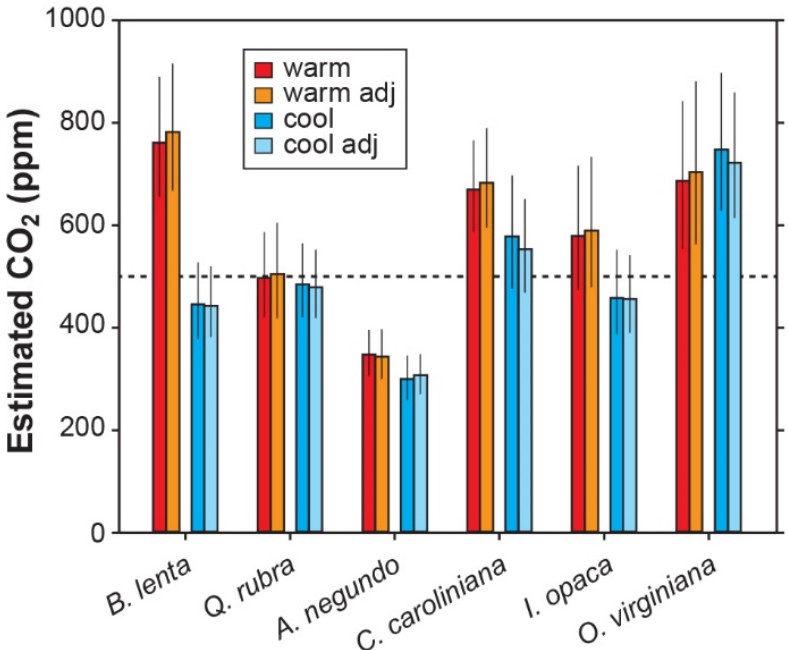

**Figure 5.** Estimates of $CO_2$ for plants grown inside growth chambers at daytime air temperatures of 28 °C and 20 °C. Also shown are estimates after taking into account the temperature sensitivity of gas diffusion ($d/v$) and the $CO_2$ compensation point in the absence of dark respiration ($\Gamma^*$) ("adj"; see also Fig. 4). Dashed line is the correct $CO_2$ concentration (500 ppm). Uncertainties in the estimates correspond to the 16th-84th percentile range.

3.3 Photorespiration

The theoretical effects of photorespiration do not strongly impact estimates of $CO_2$ in the Franks model. The average effect for our 409 extant and fossil leaves is to increase estimated $CO_2$ by 2.2% plus 38 ppm (Fig. 6). At 1000 ppm, for example, estimates would increase by 60 ppm. This calculation assumes a photorespiration fractionation ($f$) of 9.1‰, which is the value estimated for *Arabidopsis thaliana* (Schubert and Jahren, 2018). If a fractionation towards the upper bound of published estimates is used instead (15‰), estimated $CO_2$ increases on average by 3.8% plus 61 ppm. Across this range in $f$, the associated uncertainty in estimated $CO_2$ is well within the method's overall precision (~+35/-25% at 95% confidence; Franks et al., 2014). As such, $CO_2$ estimates made without these photorespiration effects (i.e. using Eq. 9 instead of Eq. 10) should be reliable, although some improvement is possible using Eq. 10 in cases where $f$ is accurately known.





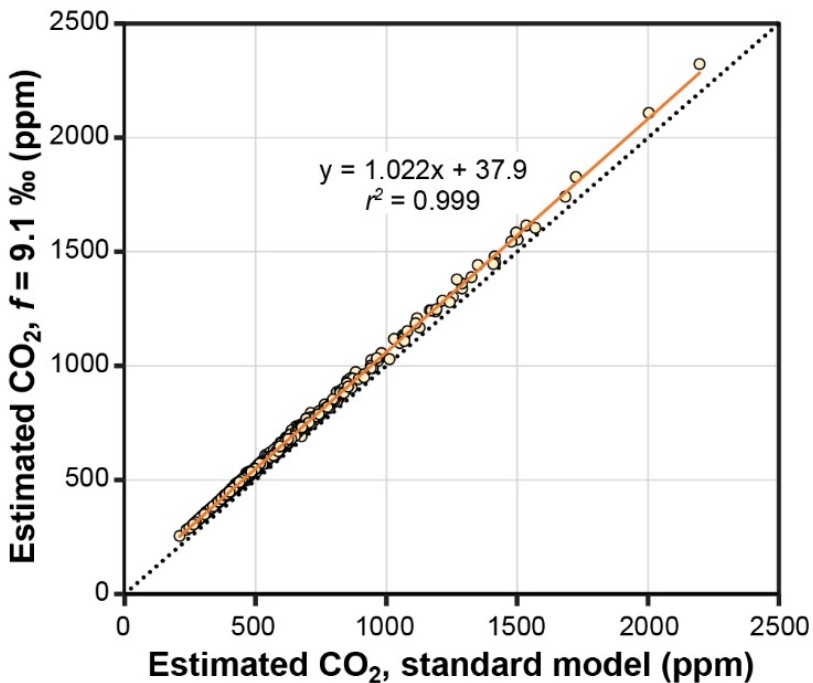

**Figure 6.** Estimates of $CO_2$ with and without a photorespiration effect ($f$ = 9.1‰; see Eq. 10). Each
symbol is an extant or fossil leaf. Dashed line is y=x.
We note that both $f$ and $\Gamma^*$ are also affected by atmospheric $O_2$ concentration. Because $O_2$ is
directly responsible for photorespiration, $f$ should scale with $O_2$ (or, more precisely, the $O_2$:$CO_2$ molar
ratio). Unfortunately, this effect is poorly constrained (Beerling et al., 2002; Berner et al., 2003; Porter et
al., 2017). In contrast, the theoretical effect of $O_2$ on $\Gamma^*$ is known: it is linear with a slope of 0.5 (Farquhar
et al., 1982; see their Eq. B13). During the Phanerozoic, $O_2$ likely ranged from 10-30%, with lows during
the early Paleozoic and early Triassic, and highs during the Carboniferous to early Permian and
Cretaceous (Berner, 2009; Glasspool and Scott, 2010; Arvidson et al., 2013; Mills et al., 2016; Lenton et
al., 2018). Assuming a present-day $\Gamma^*$ of 40 ppm (at 21% $O_2$), $\Gamma^*$ would be 49 ppm at 30% $O_2$ and 29 ppm
at 10% $O_2$. Running the Franks model on our library of 409 extant and fossil leaves, we find little effect
on estimated $CO_2$: estimates are 3.3% higher on average at 30% $O_2$ than at 10% $O_2$.
3.4 Leaves that grow close to the forest floor
$CO_2$ estimates for tropical understory leaves from five species at San Lorenzo, Panama, are very high,
ranging from 1903 to 18863 ppm (species mean = 6837 ppm). For two of the species Londoño et al.
(2018) also analyzed canopy leaves from trees nearby, and these within-species comparisons highlight
the vast discrepancy (*Ocotea* sp.: 541 vs. 5737 ppm; *Tontelea* sp.: 622 vs. 18863 ppm). The primary
difference in the inputs between the canopy and understory leaves is the $\delta^{13}C_{leaf}$: Londoño et al. (2018)
report a species-mean $\delta^{13}C_{leaf}$ of -30.0‰ for the 21 canopy species versus -35.6‰ for the five understory
species. This difference leads to very different mean estimates of $c_i/c_a$: 0.69 for canopy leaves versus a
highly unrealistic (Diefendorf et al., 2010) 0.93 for understory leaves.





It is likely that the high $CO_2$ estimates from understory leaves are mostly driven by falsely high
$\delta^{13}C_{air}$ inputs. Following the mixing model strategy outlined in Sect. 2.4 (and based on a soil organic
matter $\delta^{13}C$ of -28.2‰ measured at San Lorenzo), we calculate a species-mean $\delta^{13}C_{air}$ of -16.7‰ (mean
of intersection points in Fig. 7). When this $\delta^{13}C_{air}$ is used to estimate $CO_2$ with the Franks model (instead
of -8.5‰), the species mean drops to 699 ppm. This is somewhat higher than the species mean from
canopy leaves in the same forest (563 ppm; red triangles in Fig. 2; Londoño et al., 2018).

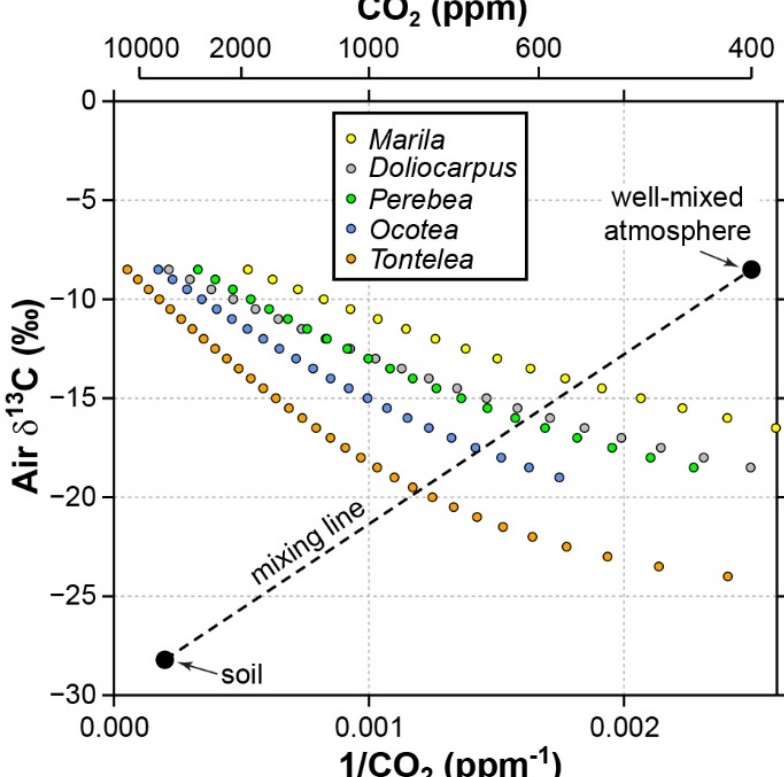

**Figure 7.** Sensitivity of estimated $CO_2$ in the Franks model to the $\delta^{13}C$ of atmospheric $CO_2$. Estimates
come from leaves of five angiosperm species that grew close to the forest floor in Parque Nacional San
Lorenzo, Panama. The step in $\delta^{13}C_{air}$ between estimates is 0.5‰. The dashed line is a two-endmember
mixing model for $CO_2$ between the soil and well-mixed atmosphere. The intersection between the
mixing model and the Franks model should correspond to the $CO_2$ concentration and $\delta^{13}C_{air}$ of the
forest-floor microenvironment.
Understory leaves from Connecticut temperate forests show similar but less dramatic patterns,
which we attribute to a more open canopy with stronger atmospheric mixing. $CO_2$ estimates for the 15
species range from 447 to 1567 ppm (mean = 794 ppm). Our intersection method identifies a mean
$\delta^{13}C_{air}$ of -11.2‰ for the Wesleyan and Connecticut College campuses (based on a soil $\delta^{13}C$ of -27.6‰
measured at Connecticut College) and -10.3‰ for Reed Gap (soil $\delta^{13}C$ = -26.4‰). Using these adjusted





$\delta^{13}C_{air}$, the species mean of estimated $CO_2$ drops to 566 ppm, which is somewhat higher than the species
mean from canopy leaves in the same areas (449 ppm; blue circles in Fig. 2).
We acknowledge that this analysis is too simple: other factors probably contribute to the
differences in estimated $CO_2$ between canopy and understory leaves. Our predicted $\delta^{13}C_{air}$ values are too
low (~8‰ and 2‰ lower than the well-mixed atmosphere for the tropical and temperate forests) and
our estimated $CO_2$ too high (~100 ppm higher than that from canopy leaves). In the lowermost 1-2
meters of the canopy, previous work suggests up to a -3‰ and +70 ppm deviation in tropical forests and
-1‰ / +20 ppm in temperate forests (Table 1). One input that could help to resolve this discrepancy is
the assimilation rate ($A_0$). We assumed a fixed $A_0$ of 12 μmol m$^{-2}$ s$^{-1}$ for all leaves, regardless of canopy
position. Shade leaves often have lower assimilation rates than sun leaves (Givnish, 1988). Substituting
lower $A_0$ values for understory leaves would lower estimated $CO_2$ roughly in proportion (Eqs. 2-3). Using
lower $A_0$ values for shade leaves in the model is appropriate, but determining the best value is difficult.
Typical $A_0$ values for leaves growing at the top of the canopy in full sun are far more consistent because
photosynthesis in these leaves is usually at its maximum capacity (saturated at full sunlight) for the
prevailing atmospheric $CO_2$ concentration. Because the degree of shadiness near the forest floor is
highly variable, photosynthesis ($A_0$) in these leaves will be acclimated to some fraction of the full-sun
maximum in a sun exposed leaf, but careful thought must go into determining what this fraction is.
We note that our mixing-model strategy cannot be applied to fossils because the global
atmospheric $CO_2$ concentration is needed (one endpoint for dashed line in Fig. 7). Instead, our
motivation for the analysis is to demonstrate that: 1) leaves growing in the lowermost 2 m of the canopy
should be considered with caution in the context of the Franks model; and 2) the failure of the model is
due to faulty inputs (mostly $\delta^{13}C_{air}$), not the model itself.
In most fossil leaf deposits, shade morphotypes are comparatively rare (e.g., Kürschner, 1997;
Wang et al., in press) because—relative to sun leaves—they are not as tough, do not travel as far by
wind, and are produced at a slower rate (Dilcher, 1973; Roth and Dilcher, 1978; Spicer, 1980; Ferguson,
1985; Burnham et al., 1992). Our recommendation is to exclude such leaves. There are several ways to
differentiate sun vs. shade morphotypes: overall shape (Talbert and Holch, 1957; Givnish, 1978;
Kürschner, 1997; Sack et al., 2006), shape of epidermal cells (larger and with a more undulated outline in
shade leaves; Kürschner, 1997; Dunn et al., 2015), vein density (lower in shade leaves; Uhl and
Mosbrugger, 1999; Sack and Scoffoni, 2013; Crifò et al., 2014; Londoño et al., 2018), and range in $\delta^{13}C_{leaf}$
(high when both sun and shade leaves are present, for example in our study; Graham et al., 2014). Not
all shade leaves grow within 2 m of the forest floor, but excluding all such leaves would eliminate the
forest-floor bias.
**4 Conclusions**
The Franks model is reasonably accurate (~28% error rate) even when the physiological inputs $A_0$
(assimilation rate at a known $CO_2$ concentration) and $g_{c(op)}/g_{c(max)}$ (ratio of operational to maximum leaf
conductance to $CO_2$) are inferred, not measured. Accuracy does improve when these inputs are
measured (~19% error rate), but such measurements are not possible with fossils and may not always
be feasible with nearest living relatives. A 28% error rate is broadly in line with (or better than) other
leading paleo-$CO_2$ proxies.
Most of the possible confounding factors that we investigated appear minor. The temperature
sensitivities of $d/v$ (related to gas diffusion) and $\Gamma^*$ ($CO_2$ compensation point in the absence of dark
respiration) have a negligible impact on estimated $CO_2$. Our temperature experiments in growth
chambers point to larger differences in some species, which must be related to incorrect values for
inputs that were not directly measured, such as $A_0$, $g_{c(op)}/g_{c(max)}$, and $g_m$ (mesophyll conductance).





Overall, though, we find that the differences in estimated $CO_2$ imparted by temperature are generally
smaller than the overall 28% error rate.
Incorporating the covariance between $CO_2$ concentration and photorespiration leads to only
small changes in estimated $CO_2$. $O_2$ concentration affects photorespiration and thus may confound $CO_2$
estimates from the Franks model, but presently the effect is poorly quantified. The effect of $O_2$ on $\Gamma^*$ is
better known, and imparts only small changes in estimated $CO_2$ across a feasible range in Phanerozoic
$O_2$ of 10-30%.
Leaves from the lowermost 1-2 m of the canopy experience slightly elevated $CO_2$ concentrations
and lower air $\delta^{13}C$ during the daytime relative to the well-mixed atmosphere. We find that if we use the
well-mixed air $\delta^{13}C$ to estimate $CO_2$ from leaves that grew near the forest floor, estimates are too high,
especially in dense tropical canopies. When we use a two-endmember mixing model to calculate the
correct local air $\delta^{13}C$, the falsely-high $CO_2$ estimates largely disappear. For fossil applications, shade
leaves from the bottom of the canopy should be avoided. Shade leaves are typically rare in the fossil
record (relative to sun leaves), and can be identified by their overall shape, the shape of their epidermal
cells, their low leaf $\delta^{13}C$, and their low vein density.
Conceptually, the Franks model holds considerable promise for quantifying paleo-$CO_2$: it is
mechanistically grounded and can be applied to most fossil leaves. Our tests of the model's accuracy
and sensitivity to temperature and photorespiration largely uphold this promise.

**Author contribution.** DR, KM, MM, and LL designed and conducted the experiments; all authors
interpreted the data; DR prepared the manuscript with contributions from all co-authors.
**Competing interests.** The authors declare that they have no conflict of interest.
**Acknowledgements.** We thank G. Dreyer and P. Siver for logistical support at Connecticut College, S.
Wang for lab assistance, and C. Crifò and A. Baresh for collecting the tropical samples. Support for LL
was provided by the Smithsonian Tropical Research Institute; the Mark Tupper Fellowship; National
Science Foundation grants EAR 0824299 and OISE, EAR, DRL 0966884; the Anders Foundation; and the
Gregory D. and Jennifer Walston Johnson and 1923 Fund.

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
