# Peer review of "$1 \qquad \text{Sensitivity of a leaf gas-exchange model for estimating paleoatmospheric CO_2}$"

_Climate of the Past, 2018_

## Referee Comment (RC1) · Anonymous Referee #1 · 16 Dec 2018

General comments: Fossil leaf gas-exchange based CO2 models are currently going through the "rigorous testing" phase and as the authors of this paper point out, this mechanistically, rather than empirically calibrated proxy, shows considerable promise. It is therefore of high relevance that studies, such as this one, are presented that provide quantification of potential confounding factors. In this case, the authors test three potential confounding factors (photorespiration, leaf temperature and canopy position) and provide quantifications on how these factors influence final CO2 estimates. They are capable of eliminating two of these factors as insignificantly affecting CO2 estimates (photorespiration and leaf temperature). The third factor, canopy position, is determined to strongly skew CO2 estimates, but the authors point out that it is possible to

identify leaves that grew in lower canopy positions, based on leaf micromorphology and an uncharacteristically wide $\delta$13C range. This paper is a relevant contribution towards quantification of the potential error in fossil leaf gas-exchange based CO2 models, and apart from minor suggested amendments, I have no problem with seeing this study being published.

Specific comments: In the materials and methods section, the authors lay out the specific ways that they are testing modern plants for potential bias in reconstructed CO2. In the appendix all the specific plants are listed with their input values and reconstructed CO2. However, from reading the methods section I get the impression that not each plant is being tested for the same potential confounding variable (photorespiration, leaf temperature and canopy position). It would be very helpful if there was a table that outlines specifically which plants were tested for what, or at least that this was made clear in the appendix, because in the main body of text it is hard to follow.

In several places in the manuscript, including the abstract, it is mentioned that the random error propagation of the Franks et al. gas exchange model is better than uncertainty estimates of other leading paleo-CO2 proxies. It would be very helpful for the untrained reader to see some proof of this statement in the form of a table that lists 1) the different CO2 proxies, 2) a method of error quantification, 3) the actual amount of uncertainty in those CO2 proxies and 4) the references to the case studies where this was tested. Such a table would lend credibility to the statement that gas-exchange models are quantifiably better than other CO2 proxies.

Final specific comment is on the title itself, for which I would like to suggest that the authors include what specifically is being tested. I.e. "Sensitivity of . . .. CO2 concentration to x, y & z". There are other variables that the model is sensitive to and I believe the title would be more informative if the specifics were included.

Technical corrections: I could not find any spelling or styling errors in the manuscript. The paper is very well constructed and easy to follow.

---

## Referee Comment (RC2) · Jennifer Mc Elwain (Referee) · 13 Feb 2019

The authors present a sensitivity analysis of a mechanistic model (Franks model) to predict paleoatmospheric CO2. They explore several specific areas; the effect of gc(op)/gc(max), A0, temperature, photorespiration and leaf canopy position on the accuracy of CO2 estimates produced by the model. In doing so, the paper adds clarity, certainty or recommendations to the model for fossil application, all of which are important additions, especially as this model is being using in a growing number of research projects. Although the paper is an important contribution, it would benefit from clarity or expansion in certain areas: 1) Aims, methods and appendix: The aims and meth-

ods section is hard to follow. This may be due to the fact the aims and rationale are mixed in with the methods. It is unclear from the text or appendix data whether all or a subset of the data is being used for each of the analysis performed. A summarised table in the methods section containing the information on the analysis being performed, data source and parameters used or tested would be beneficial (i.e. a summary of the methods in tabular format). Similarly, in the appendix, additional information on the origin of the data, sample number per species, which data points/values are measured vs estimated/assumed and a direct comparison of measured vs model estimated CO2 would greatly improve clarity.

2) Statistical analysis: Accuracy was evaluated by the degree of error rate. These claims can be strengthened by using statistical analysis. How well the model predicts CO2 could be assessed by whether or not the estimates are statistically significant different (or hopefully not) from measured CO2 values.

3) gc(op)/gc(max) and A0 (section 3.1): This section gives details about when both gc(op)/gc(max) and A0 values are either known or values from Franks et al. 2014 are used, but it would be nice to see these two parameters evaluated separately i.e. how much does gc(op)/gc(max) alone improve estimates and the same for A0. Does one contribute more than the other for improving error rates?

Additional comments: Line 86. Sensitivity saturates for some but not all taxa. See Haworth et al 2011. Line 93. A Nearest living relative or equivalent approach also get around the issue of extinct taxa. Line 156. Alternative approaches for fossils have been suggested such as estimating fossil A0 using scaling relationships between vein distance and assimilation rate however they are not discussed here (EG Montanez et al., 2016). Introduction – general comment. Critical published assessments of the Franks model are not cited (eg McElwain et al. 2016) yet they raise issues associated with parametrization of A0 and the insensitivity of CO2 estimates to variation in gamma star values which are both important discussion points in this manuscript in lines 454 -456 and 497-499. Paragraph 201-217: A some information is missing here:

chamber model/make, duration plants were grown in the chamber, light levels. What were measured vs set chamber conditions for temperature, light and CO2 (i.e. similar to how humidity is reported) Lines 232: Stomatal density/stomatal measurements and leaf stable carbon isotopes were performed on the same leaves. Clarify how this was partitioned, e.g. was the leaf divided into 2 or was a whole punch used for carbon isotopes, etc.? Lines 235: As Milligan et al is in review, I suggest adding more detail here on how $\delta$13Ca of chamber CO2 was calculated. $\delta$13Ca values of supplemented CO2 can be very negative and can vary between cylinders, unless the CO2 gas has a specific $\delta$13Ca. What is the capacity of these cylinder, in L? Figure 1: Does this need to be on a log scale? 1000 or 2000ppm are not very high values and the log scale visually skews data and error bars. A difference plot between measured and estimates plotted on a non-log scale would improve this figure. Line 351: Please provide supporting data for this statement in tabular form. What are the error rates of other proxies? Line 355: Might be helpful to report standard deviation of CO2 estimates, here and throughout the text. Line 411 to 413. Reporting of the difference between estimated and measured CO2 here is incomplete. Only means of all species investigated are provided rather than species-based diffeences or errors. For some species the error is substantial whereas other taxa show very small errors. Line 454 to 456. This supports the findings of McElwain et al 2016 Paleo 3 but it is not cited. "This compensation point ($\Gamma$âĄŐ in Eq. (2) is temperature, species and O2 dependent (Ethier and Livingston, 2004) but Franks et al. (2014) account only for the temperature dependency in the new paleo-CO2 proxy model. Allowing $\Gamma$âĄŐ to vary in response to prevailing paleoatmospheric O2 concentration [O2] ($\Gamma$âĄŐ = 1.78 × [O2]), which is known to have varied widely (10% to 30%) through the Phanerozoic (Bergman et al., 2004; Belcher and McElwain, 2008; Berner, 2009),would increase the precision of paleo-CO2 estimates but only fractionally."

Lines 500 to 506. A number of papers have suggested methods of estimating A0 to improve the accuracy of CO2 estimates using the Franks model but they are not discussed. This section would provide a good opportunity to discuss the proposed

ideas and solutions.

Section 3.4: Have any values for $\delta$13Ca been measured or are all calculated for this section? Is there any data set (from the literature or otherwise) this could be compared to? i.e. a dataset where known $\delta$13Ca is compared to itself when calculated as per the manuscript? This would strengthen this section. If $\delta$13Ca has only been calculated/inferred for this section without a comparison to measured $\delta$13Ca I think claims on the effect of $\delta$13Ca (or low canopy plants) on the model should be softened. Appendix: The authors used both known and general values for gc(op)/gc(max) and A0 to evaluate error rates but no measured values of either gc(op)/gc(max) or A0 are given in the appendix or text.

—————————————————————

---

## Author Comment (AC1) · 27 Feb 2019

We thank the reviewers for their helpful reviews; the manuscript is stronger because of them. The comments are below, followed (in bold) our responses. All line numbers by us reference the marked-up copy of the revised manuscript.

*REVIEWER #1*

General comments: Fossil leaf gas-exchange based CO2 models are currently going through the "rigorous testing" phase and as the authors of this paper point out, this mechanistically, rather than empirically calibrated proxy, shows considerable promise. It is therefore of high relevance that studies, such as this one, are presented that provide quantification of potential confounding factors. In this case, the authors test three potential confounding factors (photorespiration, leaf temperature and canopy position) and provide quantifications on how these factors influence final CO2 estimates. They are capable of eliminating two of these factors as insignificantly affecting CO2 estimates (photorespiration and leaf temperature). The third factor, canopy position, is determined to strongly skew CO2 estimates, but the authors point out that it is possible to identify leaves that grew in lower canopy positions, based on leaf micromorphology and an uncharacteristically wide δ13C range. This paper is a relevant contribution towards quantification of the potential error in fossil leaf gas-exchange based CO2 models, and apart from minor suggested amendments, I have no problem with seeing this study being published.

Specific comments: In the materials and methods section, the authors lay out the specific ways that they are testing modern plants for potential bias in reconstructed CO2. In the appendix all the specific plants are listed with their input values and reconstructed CO2. However, from reading the methods section I get the impression that not each plant is being tested for the same potential confounding variable (photorespiration, leaf temperature and canopy position). It would be very helpful if there was a table that outlines specifically which plants were tested for what, or at least that this was made clear in the appendix, because in the main body of text it is hard to follow.

**We now include this information in column E of the supplemental table.**

In several places in the manuscript, including the abstract, it is mentioned that the random error propagation of the Franks et al. gas exchange model is better than uncertainty estimates of other leading paleo-CO2 proxies. It would be very helpful for the untrained reader to see some proof of this statement in the form of a table that lists 1) the different CO2 proxies, 2) a method of error quantification, 3) the actual amount of uncertainty in those CO2 proxies and 4) the references to the case studies where this was tested. Such a table would lend credibility to the statement that gas-exchange models are quantifiably better than other CO2 proxies.

**There are of course two elements of uncertainty: precision (spread of possible solutions) and accuracy (comparison to true answer; can only be quantified for times when CO2 has been measured). The abstract brings up the theme of accuracy (28% mean error rate). In the main text (section 3.1), the mean error rate is compared generally to that in other CO2 proxies by referencing the summary work of Franks et al. (2014).**

**The error propagation scheme noted by the reviewer is related to precision. We only mention precision in the Introduction by referencing what others have found (Franks et al., 2014). It is not a focal point of the current study.**

**The reviewer may (also) be referencing the paragraph in the Introduction where we argue that studies using other stomatal-based proxies probably overstate the accuracy and precision of their CO2 estimates (lines 98-106). Our arguments here are conceptual only—there are no data we can summarize in a table, unfortunately. The point we are trying to make is that the reported accuracies and precisions associated with these other methods—when applied to plants living today (not fossils)—are better than what we find with gas-exchange methods. But this is partly because these other methods are based on empirical calibrations with…present-day plants. So excellent accuracies and precisions are not particularly surprising. But when you apply these other methods to fossils that are millions of years old, the present-day empirical calibrations are likely less appropriate.**

Final specific comment is on the title itself, for which I would like to suggest that the authors include what specifically is being tested. I.e. "Sensitivity of . . .. CO2 concentration to x, y & z". There are other variables that the model is sensitive to and I believe the title would be more informative if the specifics were included.

**The largest block of data (40 species) is "general" testing, that is, estimating $CO_2$ from field-grown trees without isolating any single confounding factor (summarized in Figure 2). Thus, it would not be fully representative to say that we were only testing the model for the influence of canopy position, temperature, and photorespiration.**

Technical corrections: I could not find any spelling or styling errors in the manuscript. The paper is very well constructed and easy to follow.

*REVIEWER #2*

The authors present a sensitivity analysis of a mechanistic model (Franks model) to predict paleoatmospheric CO2. They explore several specific areas; the effect of gc(op)/gc(max), A0, temperature, photorespiration and leaf canopy position on the accuracy of CO2 estimates produced by the model. In doing so, the paper adds clarity, certainty or recommendations to the model for fossil application, all of which are important additions, especially as this model is being using in a growing number of research projects. Although the paper is an important contribution, it would benefit from clarity or expansion in certain areas:

1) Aims, methods and appendix: The aims and methods section is hard to follow. This may be due to the fact the aims and rationale are mixed in with the methods. It is unclear from the text or appendix data whether all or a subset of the data is being used for each of the analysis performed. A summarised table in the methods section containing the information on the analysis being performed, data source and parameters used or tested would be beneficial (i.e. a summary of the methods in tabular format). Similarly, in the appendix, additional information on the origin of the data, sample number per species, which data points/values are measured vs estimated/assumed and a direct comparison of measured vs model estimated CO2 would greatly improve clarity.

**We now present a tabular summary of our study design (new Table 1).**

**In the Supplemental Table 1, we now give the sample size (column F), the target (i.e., correct) CO2 concentration (column G), and whether the input was measured or inferred (color coding of column headers). And column E gives what part of the study was addressed (general testing, temperature, or canopy position; reviewer #1 also asked for this information). We are not sure what is meant by "additional information on the origin of the data" beyond what is listed in column A and stated in the main-text Methods.**

2) Statistical analysis: Accuracy was evaluated by the degree of error rate. These claims can be strengthened by using statistical analysis. How well the model predicts CO2 could be assessed by whether or not the estimates are statistically significant different (or hopefully not) from measured CO2 values.

**We have added information about whether individual estimates depart from the target CO2 concentrations (lines 344-346 and 419-421).**

3) gc(op)/gc(max) and A0 (section 3.1): This section gives details about when both gc(op)/gc(max) and A0 values are either known or values from Franks et al. 2014 are used, but it would be nice to see these two parameters evaluated separately i.e. how much does gc(op)/gc(max) alone improve estimates and the same for A0. Does one contribute more than the other for improving error rates?

**We have added this information (lines 351-352).**

Additional comments:
Line 86. Sensitivity saturates for some but not all taxa. See Haworth et al 2011.

**We have added the qualifier "in many species".**

Line 93. A Nearest living relative or equivalent approach also get around the issue of extinct taxa.

**This is true for the stomatal ratio method, but these CO2 estimates are not meant to be quantitative in the same manner as estimates from the "full calibration" methods or the gas-exchange methods (as noted in the previous paragraph).**

Line 156. Alternative approaches for fossils have been suggested such as estimating fossil A0 using scaling relationships between vein distance and assimilation rate however they are not discussed here (EG Montanez et al., 2016).

**We have added a citation to the Montanez paper**

Introduction – general comment. Critical published assessments of the Franks model are not cited (eg McElwain et al. 2016) yet they raise issues associated with parametrization of A0 and the insensitivity of CO2 estimates to variation in gamma star values which are both important discussion points in this manuscript in lines 454 -456 and 497-499.

**As per a later comment, we have added a citation to McElwain et al. 2016 regarding gamma star on line 466.**

**Our study does not focus on the parameterization of A0, and so the associated literature does not seem relevant to the Introduction. Our study focuses on temperature, photorespiration, canopy position, as well as a general and broad test of the method.**

Paragraph 201-217: A some information is missing here: chamber model/make, duration plants were grown in the chamber, light levels. What were measured vs set chamber conditions for temperature, light and CO2 (i.e. similar to how humidity is reported)

**Chamber make/model (lines 212-213) and duration of experiment (line 229) are given. We have added information about light intensity as well as the standard deviations for temperature and CO2 concentrations in lines 213-218.**

Lines 232: Stomatal density/stomatal measurements and leaf stable carbon isotopes were performed on the same leaves. Clarify how this was partitioned, e.g. was the leaf divided into 2 or was a whole punch used for carbon isotopes, etc.?

**We now clarify our methodology in lines 237-238. We used either a hole punch or razor to remove two adjacent sections of leaf tissue near the leaf centers, avoiding major veins.**

Lines 235: As Milligan et al is in review, I suggest adding more detail here on how δ13Ca of chamber CO2 was calculated. δ13Ca values of supplemented CO2 can be very negative and can vary between cylinders, unless the CO2 gas has a specific δ13Ca. What is the capacity of these cylinder, in L?

**This paper is likely to be "in press" soon; we have appended it to the end of this file (after the marked-up copy of our manuscript). In short, a mixing line was established based on direct d13C measurements of lab air, chamber air, and cylinder CO2 (= pure CO2). We were fortunate that the d13C of the cylinder was close to the well-mixed atmosphere (the d13C in most cylinders we have used in other experiments is much more depleted). We used only the single cylinder for the duration of the experiment. The target CO2 concentration (500 ppm) was not much higher than the CO2 concentration inside the lab (~440 ppm), so we did not use much CO2.**

Figure 1: Does this need to be on a log scale? 1000 or 2000ppm are not very high values and the log scale visually skews data and error bars. A difference plot between measured and estimates plotted on a non-log scale would improve this figure.

**We prefer a log scale because it is easier to differentiate estimates at the low-end of the CO2 scale, and because the uncertainties scale in a logarithmic fashion.**

Line 351: Please provide supporting data for this statement in tabular form. What are the error rates of other proxies?

**This information was summarized by Franks et al. (2014), so we prefer not to repeat it here.**

Line 355: Might be helpful to report standard deviation of CO2 estimates, here and throughout the text.

**We now report the range that encompasses two-thirds of all estimates (lines 343-344). (Because the individual estimates are not normally distributed (tail at the high end), reporting a standard deviation can be misleading.)**

Line 411 to 413. Reporting of the difference between estimated and measured CO2 here is incomplete. Only means of all species investigated are provided rather than species-based diffeences or errors. For some species the error is substantial whereas other taxa show very small errors.

**As per an earlier comment, we now report the species-level differences on lines 419-421; no individual species-level test was significant (line 408).**

Line 454 to 456. This supports the findings of McElwain et al 2016 Paleo 3 but it is not cited. "This compensation point ($\Gamma$ * in Eq. (2) is temperature, species and O2 dependent (Ethier and Livingston, 2004) but Franks et al. (2014) account only for the temperature dependency in the new paleo-CO2 proxy model. Allowing $\Gamma$ * to vary in response to prevailing paleoatmospheric O2 concentration [O2] ($\Gamma$* = 1.78 × [O2]), which is known to have varied widely (10% to 30%) through the Phanerozoic (Bergman et al., 2004; Belcher and McElwain, 2008; Berner, 2009), would increase the precision of paleo-CO2 estimates but only fractionally."

**We have added a citation to McElwain et al. (2016 Palaeo3) (line 466).**

Lines 500 to 506: A number of papers have suggested methods of estimating A0 to improve the accuracy of CO2 estimates using the Franks model but they are not discussed. This section would provide a good opportunity to discuss the proposed ideas and solutions.

**This section deals with living leaves, where A could be measured directly. Measuring A wasn't part of our study design, unfortunately. In this section we are discussing possible reasons for noise in our mixing-model calculations. With regards to fossils, we are not recommending that our mixing model be used (line 520: "We note that our mixing-model strategy cannot be applied to fossils because…"), so the question of how to constrain A in fossils within the context of the mixing model is moot. Our take-home message for fossil applications is to avoid shade leaves (line 528), and we provide specific measurements that can be made on fossils to make this distinction, including vein density (lines 529-533).**

Section 3.4: Have any values for δ13Ca been measured or are all calculated for this section? Is there any data set (from the literature or otherwise) this could be compared to? i.e. a dataset where known δ13Ca is compared to itself when calculated as per the manuscript? This would strengthen this section. If δ13Ca has only been calculated/inferred for this section without a comparison to measured δ13Ca I think claims on the effect of δ13Ca (or low canopy plants) on the model should be softened.

**We made no direct measurements of understory d13Ca (multiple measurements over a growing season, and at different daytime hours, would be needed to calculate a representative mean value). As the reviewer correctly notes, we instead are assuming a well-behaved two end-member mixing model. We have added a note of caution related to this on lines 502-505.**

Appendix: The authors used both known and general values for gc(op)/gc(max) and A0 to evaluate error rates but no measured values of either gc(op)/gc(max) or A0 are given in the appendix or text.

**The Appendix summarizes all new data presented in the study (with the key graphics being Figures 2, 5, and 7). For these data, we \*only\* used "default" values of gop/gmax and Ao; that is, we did not measure these inputs on our leaves. As noted in the Introduction, this was a purposeful strategy because we wanted to test the CO2 model in a manner that would be similar to how most (but not all) folks will be applying the model to fossils. A "worst-case" test, if you will.**

**In the Introduction, we do summarize some of the already-published data (Figure 1). For these estimates, either gop/gmax or A0 were measured, and in most cases both were measured (lines 142-145). These data are not in the Appendix because they are already published and are not central to our study.**

**As the reviewer noted, we did additionally "degrade" these estimates by re-doing them assuming default values for gop/gmax and A0. We did this so we could compare them more directly to our estimates (lines 349-351).**

[revised manuscript text omitted]

[4]Department of Biology, Texas State University, San Marcos, TX, USA.

Corresponding author: Joseph Milligan (Joseph_Milligan@baylor.edu)

**Key Points:**

• Understanding atmospheric $CO_2$ across the Cretaceous-Paleogene boundary has been
limited due to deficiencies in existing records

• Our study highlights the utility of a proxy based on leaf gas-exchange principles

• We record a small transient rise in atmospheric $CO_2$ that is more in line with modeled
estimates of both Deccan volcanism and a bolide impact

**Abstract**

Currently there is only one paleo-$CO_2$ record from plant macrofossils that has sufficient stratigraphic resolution to potentially capture a transient spike related to rapid carbon release at the Cretaceous-Paleogene (K-Pg) boundary. Unfortunately, the associated measurements of stomatal index are off-calibration, leading to a qualitative interpretation of >2300 ppm $CO_2$. Here we re-evaluate this record with a paleo-$CO_2$ proxy based on leaf gas-exchange principles. We also test the proxy with three living species grown at 500 and 1000 ppm $CO_2$, including the nearest living relative of the K-Pg fern, and find a mean error rate of ~22%, which is comparable to other leading paleo-$CO_2$ proxies. Our fossils record a ~250 ppm increase in $CO_2$ across the K-

Pg boundary from ~625 to ~875 ppm. A small $CO_2$ spike associated with the end-Cretaceous mass extinction is consistent with many temperature records and with carbon cycle modeling of

Deccan volcanism and the meteorite impact.

**Plain Language Summary**

Currently there is only one paleo-$CO_2$ record close enough to the Cretaceous-Paleogene (K-Pg) boundary to record a rapid release in atmospheric $CO_2$, a greenhouse gas. This record is based on the stomatal frequencies of fern fossils at the K-Pg boundary and *Ginkgo* fossils before and after the boundary. Unfortunately, due to deficiencies with the method, the $CO_2$ inferences are only qualitative. Here we look at the same fossils with a proxy based on leaf gas-exchange principles (i.e. photosynthesis). We first test the proxy with three living species grown at 500 and

1000 ppm $CO_2$, including the nearest living relative of the K-Pg fern, and find a comparable accuracy to other quantitative paleo-$CO_2$ proxies. The fossils record a modest ~250 ppm increase in $CO_2$ across the K-Pg boundary. These estimates are consistent with most temperature records and with carbon cycle modeling of Deccan volcanism and the meteorite impact.

**1 Introduction**

The Cretaceous–Paleogene (K-Pg) boundary ~66 Ma marks one of the largest mass extinctions in Earth's history (Alroy, 2008; Brusatte et al., 2015; McElwain and Punyasena,

2007; Raup and Sepkoski, 1982). The concentration of atmospheric $CO_2$ may have risen abruptly at this time, contributing to the biological upheaval (Beerling et al., 2002). Removal of an instantaneous release of $CO_2$ to the atmosphere typically requires up to 100-200 kyrs, following exponential decay due to silicate weathering (Archer, 2005; Colbourn et al., 2015; Schaller et al.,

2011; Zeebe and Zachos, 2013). Adequate constraints on atmospheric $CO_2$ from proxy records during this critical period have been missing, mostly because of a lack in sufficient stratigraphic resolution to definitively identify individual records occurring <100 kyrs after the extinction event. This is because either the stratigraphic section is too coarse to resolve 100 kyrs of time (Steinthorsdottir et al., 2016) or because definitive markers of the boundary (e.g., iridium spike, presence of microspherules) are missing (Huang et al., 2013; Nordt et al., 2003; Zhang et al.,

2018).

One exception is the study of Beerling et al. (2002), who used stomatal indices (SI, stomatal density normalized by the number of epidermal cells) to estimate $CO_2$ from fern macrofossils (aff. *Stenochlaena*) that occur 5-25 cm above the K-Pg boundary in the Raton

Basin, New Mexico. In this stratigraphic section, the K-Pg boundary is identified by an iridium spike and shocked quartz, and the fossils come from sediments that contain, and lie directly above, the fern spore spike. This fern spike is present across the globe (Vajda et al., 2001) and likely occurred within $10^3$ yrs after the K-Pg boundary (Clyde et al., 2016). Thus, the aff.

*Stenochlaena* fossils should record any transient rise in atmospheric $CO_2$ associated with the

Chicxulub impact and K-Pg boundary. Indeed, the fossils likely capture close to the peak in $CO_2$

change because after an instantaneous release, $CO_2$ will remain significantly elevated for hundreds of years (Solomon et al., 2009; Zeebe, 2013). Unfortunately, the measured stomatal indices fall well below the present-day calibration of *S. palustris*, leading Beerling et al. (2002)

to interpret a $CO_2$ concentration that exceeded the calibrated space (>2300 ppm), considerably higher than latest Cretaceous and earliest Paleocene $CO_2$ values of ~350-550 ppm inferred from

*Ginkgo* fossils (Beerling et al., 2002; 2009). The Beerling et al. (2002) study thus suggests a very large, but poorly constrained, $CO_2$ pulse.

Leaf gas-exchange models are an alternative to stomatal density (SD) and SI proxies for estimating paleo-$CO_2$ concentration (Franks et al., 2014; Konrad et al., 2008, 2017). The model developed by Franks et al. (2014) depends on the well-established relationship between the rate of $CO_2$ assimilation of plants (A), leaf conductance to $CO_2$ ($g_{ctot}$), and the difference between atmospheric ($c_a$) and leaf intercellular $CO_2$ ($c_i$) (Farquhar and Sharkey, 1982):

$$A = g_{ctot}(c_a - c_i) \qquad (1)$$

Equation 1 can be rearranged to solve for atmospheric $CO_2$ (Equation 2). The three input variables needed are the average assimilation rate (determined from a nearest living relative), average total leaf conductance (determined largely from SD and stomatal size measured on the fossil), and average $c_i/c_a$ (determined from the fossil leaf and air carbon isotopic composition combined with knowledge of the fractionation process) (Franks et al., 2014):

$$C_a = \frac{A}{g_{ctot}\left(1 - {c_i}/{C_a}\right)} \qquad (2)$$

The model has been used to reconstruct $CO_2$ during the Phanerozoic (Franks et al., 2014), including the late Paleozoic (Montañez et al., 2016), middle Cretaceous (Richey et al., 2018),

Late Cretaceous (Martínez et al., 2018), early Paleocene (Kowalczyk et al., 2018), middle

Eocene (Maxbauer et al., 2014; Wolfe et al., 2017), Oligocene-Miocene boundary (Reichgelt et al., 2016; Tesfamichael et al., 2017) and early Miocene (Londoño et al., 2018).

Leaf gas-exchange models provide at least five crucial advantages over other stomatal approaches: (1) they are based mechanistically on physiological principles, not empirical, species-specific calibrations; (2) measurements of SD, a component of $g_{ctot}$, are typically more reliable and easier to make than SI because epidermal cells can be difficult to count (Barclay and

Wing, 2016); (3) they are less sensitive to the saturating effect that can limit other stomatal methods to <500-1000 ppm $CO_2$ (e.g. Doria et al., 2011); (4) they can be applied to most subaerial leaves from $C_3$ species, regardless of age or taxonomy; and (5) they are not restricted to species whose SD or SI is sensitive to $CO_2$ because the models have multiple physiological inputs with well-understood sensitivities to $CO_2$. Importantly, these gas-exchange methods open up much of the paleobotanical record for quantitative $CO_2$ inference, not just to fossil taxa that are still living today. While the Franks et al. (2014) model shows promise, more extensive testing will improve confidence in the $CO_2$ estimates. Specifically, model validation with extant species has been limited to mostly angiosperms and a few gymnosperms, neglecting major clades such as ferns and lycophytes. Additionally, the model has been tested at elevated $CO_2$ on only a few species (Franks et al., 2014; Londoño et al., 2018).

Here we test the model using growth-chamber experiments at elevated $CO_2$ (500 and

1000 ppm) for two ferns (*Osmundastrum cinnamomeum* (L.) C. Presl and a close living relative to the K-Pg fern, *Stenochlaena palustris* (Burm.f.) Bedd.), and one conifer (*Cedrus deodara*

(Roxb.) Loud. We then use the same fossils of aff. *Stenochlaena* and *Ginkgo* from Beerling et al.

(2002) to re-evaluate atmospheric $CO_2$ across the K-Pg boundary using the gas-exchange model of Franks et al. (2014).

**2 Materials and methods**

For detailed methods and all data, see the supporting information.

*2.1 Growth chamber experiments*

All plants were potted with Premier Horticulture "Pro-mix" Bx with Mycorise and grown in two Conviron E7/2 growth chambers. Plants were watered to field capacity daily and given

Scotts all-purpose flower and vegetable fertilizer (10-10-10) every two months. The chamber conditions were set to a 17-hour photoperiod with a 30-minute simulated dawn and dusk.

Temperature was $25 \pm 0.2°C$ ($1\sigma$) during the day and $20 \pm 1°C$ ($1\sigma$) at night. The relative humidity was $84 \pm 5\%$ ($1\sigma$) and the $CO_2$ concentration was either $500 \pm 25$ ($1\sigma$) or $1000 \pm 15$

($1\sigma$) ppm. Growth light levels (photosynthetically active radiation, or PAR) varied between 100-

500 µmol $m^{-2}$ $s^{-1}$ depending on plant height. Plants were rotated between the two chambers every two weeks to negate any chamber effects (e.g., Porter et al., 2015).

*2.2 Fossil leaves*

The fossils come from Beerling et al. (2002). The aff. *Stenochlaena* fossils were collected at the Clear Creek South locality in the Raton Basin, New Mexico (Wolfe and Upchurch, 1987).

The fossils represent an extinct (and currently unnamed) genus related to *Stenochlaena* (Wolfe and Upchurch, 1987), with identification based on venation, tooth and frond architecture, and stomatal anatomy, especially maceration-resistant cutin lamellae on the guard cells (Beerling et al., 2002; Wolfe and Upchurch, 1987). The stratigraphic interval containing the fern fossils includes the top of the fern spore spike and the overlying level where dicot pollen returns to dominance.

The latest Cretaceous and earliest Paleocene *Ginkgo adiantoides* fossils were obtained by loan from the Denver Museum of Nature and Science (DMNH) and the Yale Peabody Museum (YPM), respectively. The Cretaceous fossils come from the Hell Creek Formation in the

Williston Basin of North Dakota (DMNH site 566), 33.5 m below the K-Pg boundary (Johnson,

2002). Based on constraints from geochronology, magnetostratigraphy, and sedimentation rates,

Hicks et al. (2002) consider the locality 0.5 Myrs older than the K-Pg boundary. The early

Paleocene fossils come from the Fort Union Formation in the Bighorn Basin of Wyoming (YPM

locality 7659), 4 m above the K-Pg boundary; based on sedimentation rates, Wing et al. (1995)

interpret the site to post-date the K-Pg boundary by 0.5 Myrs. We assume a K-Pg boundary age of 66 Ma (Gradstein et al., 2012; Renne et al., 2013).

*2.3 Leaf gas-exchange model*

The Franks et al. (2014) leaf gas-exchange model has 16 inputs that are used to calculate the average assimilation rate, total leaf conductance, and $c_i/c_a$ (Equation 2). When possible, we measured the inputs directly, including SD, stomatal pore length, single guard cell width, and leaf $\delta^{13}C$ (Table S1). For living plants, the assimilation rate, A, and operational stomatal conductance to $CO_2$, $g_{c(op)}$, were also measured with a LI-COR 6400 portable photosynthesis system. These measurements were made under environmental conditions identical (or nearly identical) to the growth chamber environment. Leaves first equilibrated inside the leaf chamber for 10 to 30 minutes. All reported results are means of the most stable individual measurements (typically <5% variance across measurements).

For the fossil leaves, nearest living relatives were used to assign taxon-specific values of

$A_0$ (assimilation rate at a known $CO_2$ concentration) and $g_{c(op)}/g_{c(max)}$ (ratio of operational to maximum stomatal conductance to $CO_2$). For aff. *Stenochlaena*, values come from *S. palustris*

reported here; for *G. adiantoides*, values come from field-grown *G. biloba* at ~400 ppm $CO_2$

(Kowalczyk et al., 2018).  For other inputs not directly measured, we used the recommended values from Franks et al. (2014) or appropriate values from the literature (see Dataset S1). To solve for atmospheric $CO_2$, we use the Kowalczyk et al. (2018) code written in R (v.3.4.4; R core team, 2018).

As with the Beerling et al. (2002) study, our atmospheric $CO_2$ reconstruction comes from two different species at three different localities. Because this potentially introduces species and environmental effects, we performed a sensitivity analysis by estimating $CO_2$ after sequentially varying each input parameter across a range typical for $C_3$ plants. Consistent with previous work (Kowalczyk et al., 2018; Maxbauer et al., 2014; McElwain et al., 2016) we find that among the inputs that cannot be measured directly on fossils, changes in $A_0$ and $g_{c(op)}/g_{c(max)}$ have the biggest impact on estimated $CO_2$ (Figure S16). As such, we explored how different value choices for these inputs may affect our $CO_2$ estimates. For example, because a one-step change in $CO_2$ may not induce the same physiological response as a slow-and-steady $CO_2$ increase over geological time, we evaluated the model both with the measured physiological inputs (discussed earlier) and generic values recommended by Franks et al. (2014) (Table S2).

We note that the Franks et al. (2014) leaf gas-exchange model is based on leaf temperature, not air temperature. Both theory (Michaletz et al., 2015; Michaletz et al., 2016) and observations (Helliker and Richter, 2008; Song et al., 2011) indicate that the control of leaf gas exchange leads to relatively stable assimilation-weighted leaf temperatures (~19-25 °C from temperate to tropical regions; i.e., thermoregulation).Thus, despite significant changes (e.g., several degrees) in global mean air temperature, as often observed across the K-Pg boundary, daytime leaf temperature during the growing season should stay relatively constant. If instead leaf temperature did vary substantially, it could have mixed effects on many model inputs (A,

$g_{c(op)}/g_{c(max)}$, SD, stomatal size, $c_i/c_a$); for example, an increase in A with no changes to other inputs will cause an equally proportional increase in estimated $CO_2$ (Figure S16B). While assimilation rates can increase with leaf temperature within seconds to hours (e.g. Berry and

Björkman, 1980); $C_3$ plants generally exhibit stable assimilation rates when acclimated to a range of growth temperatures (i.e., temperature homeostasis of photosynthesis, Yamori et al., 2014).

With regards to the Franks et al. (2014) model, tests on six species grown at 20 and 28 $^o$C air temperature show only a mild effect on the ability of the model to estimate $CO_2$ (Royer et al.,

2018). For these reasons, we argue that changes in mean global temperature probably have little impact on the reliability of our $CO_2$ reconstructions.

*2.4 Statistics*

A one-sample Kolmogorov-Smirnov test identified that most of our inputs did not have normal distributions (Dataset S1). Thus, for our experiments, we used a two-sample

Kolmogorov-Smirnov test to test for differences between $CO_2$ treatments in the inputs. All analyses were done within R and performed at the plant level.

**3 Results and Discussion**

*3.1 Growth chamber experiments*

The median $CO_2$ estimates for the three living species in the 500 ppm $CO_2$ treatment range from 584-686 ppm, and in the 1000 ppm treatment from 1016-1442 ppm (Figure 1; Table

S2). Across all species, the 500 and 1000 ppm $CO_2$ treatments have a mean error rate

[(|estimated $CO_2$-observed $CO_2$|) / (observed $CO_2$)] of ~25% and ~19%, respectively. This is higher than elevated $CO_2$ experiments of *Wollemia nobilis* at 480 and 1270 ppm (7%; Franks et al., 2014), but is comparable to other paleo-$CO_2$ proxies at present day $CO_2$ such as alkenones (12.4%; Pagani, 2002), boron isotopes (8.2%; Henehan et al., 2013; Hönisch and Hemming,

2005), and pedogenic carbonates (67%; Ekart et al., 1999). Additionally, the precision of estimates within this study are comparable or better than other paleo-$CO_2$ proxies, especially at elevated $CO_2$ (Beerling et al., 2009; Montañez et al., 2011; Royer, 2014).Using the generic values recommended by Franks et al. (2014) for $A_0$ and $g_{c(op)}/g_{c(max)}$, median $CO_2$ estimates increase for *S. palustris* and *O. cinnamomeum* while decreasing for *C. deodara*, with a mean error rate of 44% and 21% for the 500 and 1000 ppm $CO_2$ treatments (Table S2). Note, however, that the generic values recommended by Franks et al. (2104) were obtained for field conditions which may differ slightly from growth chambers.  Plants in growth chambers typically experience lower light and higher humidity, which affect $A_0$ and $g_{c(op)}/g_{c(max)}$ via $g_{c(op)}$.

  *O. cinnamomeum* and *C. deodara* show no significant differences to $CO_2$ in SD, guard cell length, stomatal pore length, single guard cell width, and $g_{c(op)}/g_{c(max)}$ (P>0.05), but both have significantly higher A at 1000 ppm $CO_2$ (P=0.03; P=0.02; Figure 2). SD in *S. palustris* declines significantly by 21% at high growth $CO_2$ (P=0.048), but with no significant change in guard cell length, stomatal pore length, or guard cell width (P>0.05). *C. deodara* and *S. palustris* exhibit a significant increase in $c_i/c_a$ at elevated $CO_2$ (P=0.004; P=0.048), while *O. cinnamomeum* does not.

  The disparate physiological and morphological responses to $CO_2$ highlight an advantage of leaf gas-exchange proxies over other stomatal proxies. If SD or SI does not respond to $CO_2$, then by definition the SD and SI methods cannot be used (see Reichgelt et al., 2016). For leaf gas-exchange models, this is not necessarily true if other inputs do respond to $CO_2$. This is in fact the case with *O. cinnamomeum* and *C. deodara*, which produced reasonable $CO_2$ estimates for both treatments despite no changes in SD. Part of the issue with the other stomatal proxies is that they depend on a calibrated response, and the timescale associated with these responses (typically months to years) may not be sufficiently long, especially at higher-than-present $CO_2$

concentrations (Royer, 2001; see multi-year response from Hincke et al., 2016).

*3.2 K-Pg boundary CO$_2$*

The leaf gas-exchange estimates of CO$_2$ from *G. adiantoides* are similar for the Late

Cretaceous (66.5 Ma; 624 ppm; 95% confidence interval 454-882 ppm) and early Paleocene (65.5 Ma; 630 ppm; 95% confidence interval 408-1181 ppm) (Figure 3; Table S2). The larger uncertainty with the Paleocene estimate is mostly due to having to model both stomatal pore length and single guard cell width because we were unable to measure them (Table S1). The leaf gas-exchange estimate of CO$_2$ from aff. *Stenochlaena* directly after the K-Pg boundary is 873

ppm (95% confidence interval 550-1414 ppm). By comparison, the estimates from Beerling et al.

(2002) (updated by Beerling et al., 2009) are 539 ppm for the Late Cretaceous, >2300 ppm for the fern layer, and 343 ppm for the early Paleocene.

It is possible that all three of our estimates are falsely-high because the model overestimates present-day CO$_2$ for *G. biloba* (Barclay and Wing, 2016; Kowalczyk et al., 2018; but see Franks et al., 2014) and *S. palustris* at both 500 and 1000 ppm CO$_2$ (Figure 1). The relative temporal patterns, though, are more likely to be robust. If we use the generic inputs for

$A_0$ and $g_{c(op)}/g_{c(max)}$ recommended by Franks et al. (2014), all three estimates increase by ~200-

500 ppm (Table S2 and Figure 3), but the increase in CO$_2$ between the Late Cretaceous and fern layer does not change by very much (+250 ppm) and remains fundamentally different from the original interpretation of Beerling et al. (2002) (Figure 3).

A source of uncertainty for the aff. *Stenochlaena* CO$_2$ estimate is the atmospheric $\delta^{13}$C

directly at the K-Pg boundary, which affects the calculation of $c_i/c_a$. Measured carbon isotopic excursions at the K-Pg boundary range from 0 to -3‰ (Arens and Jahren, 2000; Beerling et al.,

2001; Maruoka et al., 2007; Schimmelmann and DeNiro, 1984; Schulte et al., 2010). Where examined in detail, the excursion in terrestrial sections begins immediately above the K-Pg boundary clay in the fern spike interval, with the most negative values in the early phase of dicot recovery, and a return to pre-excursion values no higher than 2-3 m up section (reviewed in

Upchurch et al., 2007). For our initial modeling we assume -2‰ (Text S2). If we instead assume an excursion of 0‰, comparable to the value at the top of the K-Pg boundary clay, or -3‰, the median $CO_2$ is 1170 and 762 ppm, respectively. Neither of these changed estimates strongly affect our key interpretations.

Our $CO_2$ record implies a transient change of ~+250; if we take the extreme scenario of comparing the lower and upper bounds of the 95% confidence intervals, this change could range from -333 to +1032 ppm. Critically, we provide the first fully-bounded $CO_2$ estimate from the top of the fern spike interval, and thus likely from within the first $10^3$ years after the bolide impact. Our *Ginkgo* estimates bracket the event by roughly 500 kyrs, meaning that we do not know the $CO_2$ concentration directly before the bolide impact. This is an important deficiency because global temperatures rose ~300 kyrs before the K-Pg boundary and subsequently fell leading up to the boundary (Barnet et al., 2017; Nordt et al., 2003; Petersen et al., 2016; Wilf et al., 2003; Zhang et al., 2018). Zhang et al. (2018) estimate with the pedogenic carbonate proxy a

$CO_2$ concentration of 700 ppm 110 kyrs before the K-Pg boundary (Figure S17), suggesting that

Deccan volcanism caused an elevation in $CO_2$ before the boundary (Courtillot et al., 1986; Tobin et al., 2017 and sources cited within) and therefore the $CO_2$ spike we report may not be contributed entirely by the bolide-impact.

The Chicxulub bolide impact would release $CO_2$ almost instantaneously via the vaporization of target carbonate bedrock (Artemieva and Morgan, 2017; O'Keefe and Ahrens,

1989) and wildfires (Durda and Kring, 2004; Wolbach et al., 1990). A recent model for the vaporization of target carbonate bedrock at Chicxulub suggests a modest 54 ppm rise in atmospheric $CO_2$ (Artemieva et al., 2017). Global wildfires may have caused $CO_2$ to increase by

315 ppm (Toon et al., 2016), but the extent of these fires is contentious and may have been far less (Belcher et al., 2003, 2004, 2005, 2009, 2015; Belcher, 2009; Harvey et al., 2008; Morgan et al. 2013).

 Establishing a link between Deccan volcanism and $CO_2$ change at the K-Pg boundary is difficult because: 1) age uncertainties of the lava flows are on the order of $10^4$-$10^5$ yrs (e.g.,

Renne et al., 2015; Schoene et al., 2015; Schoene et al., 2019; Sprain et al., 2019); and 2)

constraining the amount and rate of $CO_2$ release is challenging (Jay and Widdowson, 2008; Self et al., 2006). Deccan volcanism clearly brackets the K-Pg boundary, but whether there was a pulse of activity within $10^2$-$10^3$ yrs of the boundary is unresolved (Schoene et al., 2019; Sprain et al., 2019). Using existing constraints on the magnitude and pacing of $CO_2$ release for the Deccan,

Tobin et al. (2017) demonstrate that it is possible, in principle, to raise $CO_2$ concentrations by several hundred ppm. Future work may provide clarity.

 Temperature records spanning the first $10^2$-$10^3$ yrs after the K-Pg boundary are sparse, but most modeling and high resolution marine data are not consistent with a large change in $CO_2$.

After a brief "impact winter" (months to decades; Bardeen et al., 2017; Brugger et al., 2017;

Taylor et al., 2018; Vellekoop et al., 2014, 2015, 2016), temperatures increased between ~1-6 °C

depending on paleolatitude and geographic location, with the largest increases often at higher paleolatitudes (Macleod et al., 2018; Taylor et al., 2018; Vellekoop et al., 2014; Zhang et al.,

2018). Terrestrial temperature trends inferred from leaf fossils are somewhat ambiguous and model dependent (Upchurch et al., 2007). Among marine records and most relevant to our study,

Taylor et al. (2018) document a 2.5-4 °C warming during the fern spike interval in the southern mid-latitudes (present-day New Zealand). Together, these reconstructions best fit a scenario with a modest 1-3 °C rise in global mean surface temperature. If we assume an Earth-system sensitivity of 3 °C or higher per $CO_2$ doubling (Royer 2016), these records imply—at most—one

$CO_2$ doubling. One exception is a ~5 °C warming within ~100,000 yrs after the K-Pg boundary at the global stratotype El Kef, Tunisia (~20 °N paleolatitude; MacLeod et al., 2018). This subtropical temperature record appears incompatible with our record, suggesting that either $CO_2$

directly before the K-Pg boundary was substantially lower (<400 ppm) than what our and most other reconstructions imply (Zhang et al., 2018; see also Figure S17) or local changes in ocean chemistry biased the temperature estimates.

   In summary, we find no strong evidence for a large pulse of atmospheric $CO_2$ coincident with the K-Pg boundary. Our $CO_2$ record from within or directly above the fern spike is most consistent with a $CO_2$ rise of no more than ~500 ppm and more likely ~250 ppm or less. This is in keeping with the balance of evidence from temperature records and from the carbon cycle modeling of impact vaporization of target bedrock, widespread wildfire, and Deccan volcanism.

**Acknowledgements**

   We thank S. Sultan for use of her LI-COR gas-exchange analyzer, T. Ku for providing laboratory space, and D. Penman and one anonymous reviewer for helpful comments. Data used in this study is available in the Supplementary Information and DatasetSI.

**Figure 1.** Atmospheric $CO_2$ estimates and probability density function using the leaf gas- exchange model of Franks et al. (2014) with *Cedrus deodara (C.d.)*, *Osmundastrum*

*cinnamomeum (O.c.)*, and *Stenochlaena palustris (S.p.)*, grown at two $CO_2$ treatments (500 and

1000 ppm $CO_2$). Dotted lines represent the target $CO_2$ concentrations. Estimates are the median and 95% confidence interval.

**Figure 2.** Measured inputs for *Cedrus deodara (C.d.)*, *Osmundastrum cinnamomeum (O.c.)*, and

*Stenochlaena palustris (S.p.)* grown at two $CO_2$ concentrations (500 and 1000 ppm) and fossil

*Ginkgo adiantoides (G.a.)* and aff. *Stenochlaena*. Abbreviations: K, Cretaceous; Pg, Paleogene.

For multiple comparisons different letters indicate significantly different values at the 0.05 level.

* $P \le 0.05$, ** $P \le 0.01$, *** $P \le 0.001$.

**Figure 3.** Atmospheric $CO_2$ estimates from the Cretaceous-Paleogene boundary. Estimates from the leaf gas-exchange model (this study) are based on the same fossils whose stomatal index was used to estimate $CO_2$ by Beerling et al. (2002). The gray squares are based on the recommended values from Franks et al. (2014) for assimilation rate and the ratio of operational to maximum stomatal conductance. Error bars represent the 95% confidence interval.

**Figure 1.**

[Figure]

**Figure 2.**

[Figure]

**Figure 3.**

[Figure]

---

## Author Response (AR1)

We thank the editor and reviewers for their helpful reviews; the manuscript is stronger because of them. The comments are below, followed (in bold) our responses. All line numbers by us reference the marked-up copy of the revised manuscript.

*EDITOR*

Comments to the Author:
Thank you for addressing the reviewer comments. Both reviewers were positive about the paper and I also think it is an important contribution to the paleo-CO2 literature. The revisions proposed address most of their concerns. I note a few issues below that I would like to see addressed before final acceptance and ask that you submit a revised version of the paper that includes the revisions you proposed and addresses these further comments:

1) Line 98-101. Can you clarify this discussion of uncertainty? The issue is whether all studies using only stomatal density ignore the two sources of error you mention, or just some do. The wording here is not clear on that point.

**We have clarified this point (lines 99-101). Very few studies propagate both sources of uncertainty (the Beerling 2009 study, mentioned at the end of the sentence, is one such study).**

Line 102. Smaller than what? With fossil taxa? Please clarify.

**We have clarified this point (smaller than with gas-exchange proxies; line 103).**

Line 116-118. This sentence strikes me as awkward it refers to "elements" but then phrases them as questions. Can you reword?

**Done (lines 116-117)**

Line 121. Can you provide a more informative title for the table?

**Done (line 122)**

Line 170-171. How are the ambient CO2 values known?

**References added to the Mauna Loa and Harvard Forests databases (lines 171-173).**

Line 239. Regarding Milligan et al., please just provide the details of the method. Even if the paper is in press the reader would benefit.

**Done (lines 242-243). Also, the Milligan paper is now published, with a doi.**

Line 320-321. I think it would help here to explain that you do not have measurements of d13C-CO2.

**We now note this (lines 324-325); we also mention this in the results (lines 508-510)**

Line 342-346. I understand why you use the 2/3 range, but you should explain here why. Also, what does it mean to say that 28% overlap the target at 95% confidence if they are really noisy (that is, is this a useful statement). And, how is the target value established? Assumed 400 ppm?

**We now say that the 2/3 range is a close equivalent to +/- 1 standard deviation (lines 349-350). The target value is assumed to be 400 ppm (we now state this directly on line 171).**

**We agree that this is a noisy signal, and the reader can decide whether the signal:noise ratio is good enough for their purposes. We included this statement on the account of a direct request by one of the reviewers.**

Line 526. Can you replace "tough" with a more specific term?

**We replaced this word with "durable" (line 533) (toughness is a quantitative trait made by plant ecologists).**

With best wishes, Ed Brook
* * *
*REVIEWER #1*

General comments: Fossil leaf gas-exchange based CO2 models are currently going through the "rigorous testing" phase and as the authors of this paper point out, this mechanistically, rather than empirically calibrated proxy, shows considerable promise. It is therefore of high relevance that studies, such as this one, are presented that provide quantification of potential confounding factors. In this case, the authors test three potential confounding factors (photorespiration, leaf temperature and canopy position) and provide quantifications on how these factors influence final CO2 estimates. They are capable of eliminating two of these factors as insignificantly affecting CO2 estimates (photorespiration and leaf temperature). The third factor, canopy position, is determined to strongly skew CO2 estimates, but the authors point out that it is possible to identify leaves that grew in lower canopy positions, based on leaf micromorphology and an uncharacteristically wide δ13C range. This paper is a relevant contribution towards quantification of the potential error in fossil leaf gas-exchange based CO2 models, and apart from minor suggested amendments, I have no problem with seeing this study being published.

Specific comments: In the materials and methods section, the authors lay out the specific ways that they are testing modern plants for potential bias in reconstructed CO2. In the appendix all the specific plants are listed with their input values and reconstructed CO2. However, from reading the methods section I get the impression that not each plant is being tested for the same potential confounding variable (photorespiration, leaf temperature and canopy position). It would be very helpful if there was a table that outlines specifically which plants were tested for what, or at least that this was made clear in the appendix, because in the main body of text it is hard to follow.

**We now include this information in column E of the supplemental table.**

In several places in the manuscript, including the abstract, it is mentioned that the random error propagation of the Franks et al. gas exchange model is better than uncertainty estimates of other leading paleo-CO2 proxies. It would be very helpful for the untrained reader to see some proof of this statement in the form of a table that lists 1) the different CO2 proxies, 2) a method of error quantification, 3) the actual amount of uncertainty in those CO2 proxies and 4) the references to the case studies where this was tested. Such a table would lend credibility to the statement that gas-exchange models are quantifiably better than other CO2 proxies.

**There are of course two elements of uncertainty: precision (spread of possible solutions) and accuracy (comparison to true answer; can only be quantified for times when CO2 has been measured). The abstract brings up the theme of accuracy (28% mean error rate). In the main text (section 3.1), the mean error rate is compared generally to that in other CO2 proxies by referencing the summary work of Franks et al. (2014).**

**The error propagation scheme noted by the reviewer is related to precision. We only mention precision in the Introduction by referencing what others have found (Franks et al., 2014). It is not a focal point of the current study.**

**The reviewer may (also) be referencing the paragraph in the Introduction where we argue that studies using other stomatal-based proxies probably overstate the accuracy and precision of their CO2 estimates (lines 98-107). Our arguments here are conceptual only—there are no data we can summarize in a table, unfortunately. The point we are trying to make is that the reported accuracies and precisions associated with these other methods—when applied to plants living today (not fossils)—are better than what we find with gas-exchange methods. But this is partly because these other methods are based on empirical calibrations with…present-day plants. So excellent accuracies and precisions are not particularly surprising. But when you apply these other methods to fossils that are millions of years old, the present-day empirical calibrations are likely less appropriate.**

Final specific comment is on the title itself, for which I would like to suggest that the authors include what specifically is being tested. I.e. "Sensitivity of . . .. CO2 concentration to x, y & z". There are other variables that the model is sensitive to and I believe the title would be more informative if the specifics were included.

**The largest block of data (40 species) is "general" testing, that is, estimating $CO_2$ from field-grown trees without isolating any single confounding factor (summarized in Figure 2). Thus, it would not be fully representative to say that we were only testing the model for the influence of canopy position, temperature, and photorespiration.**

Technical corrections: I could not find any spelling or styling errors in the manuscript. The paper is very well constructed and easy to follow.

The authors present a sensitivity analysis of a mechanistic model (Franks model) to predict paleoatmospheric CO2. They explore several specific areas; the effect of gc(op)/gc(max), A0, temperature, photorespiration and leaf canopy position on the accuracy of CO2 estimates produced by the model. In doing so, the paper adds clarity, certainty or recommendations to the model for fossil application, all of which are important additions, especially as this model is being using in a growing number of research projects. Although the paper is an important contribution, it would benefit from clarity or expansion in certain areas:

1) Aims, methods and appendix: The aims and methods section is hard to follow. This may be due to the fact the aims and rationale are mixed in with the methods. It is unclear from the text or appendix data whether all or a subset of the data is being used for each of the analysis performed. A summarised table in the methods section containing the information on the analysis being performed, data source and parameters used or tested would be beneficial (i.e. a summary of the methods in tabular format). Similarly, in the appendix, additional information on the origin of the data, sample number per species, which data points/values are measured vs estimated/assumed and a direct comparison of measured vs model estimated CO2 would greatly improve clarity.

**We now present a tabular summary of our study design (new Table 1).**

**In the Supplemental Table 1, we now give the sample size (column F), the target (i.e., correct) CO2 concentration (column G), and whether the input was measured or inferred (color coding of column headers). And column E gives what part of the study was addressed (general testing, temperature, or canopy position; reviewer #1 also asked for this information). We are not sure what is meant by "additional information on the origin of the data" beyond what is listed in column A and stated in the main-text Methods.**

2) Statistical analysis: Accuracy was evaluated by the degree of error rate. These claims can be strengthened by using statistical analysis. How well the model predicts CO2 could be assessed by whether or not the estimates are statistically significant different (or hopefully not) from measured CO2 values.

**We have added information about whether individual estimates depart from the target CO2 concentrations (lines 350-352 and 425-427).**

3) gc(op)/gc(max) and A0 (section 3.1): This section gives details about when both gc(op)/gc(max) and A0 values are either known or values from Franks et al. 2014 are used, but it would be nice to see these two parameters evaluated separately i.e. how much does gc(op)/gc(max) alone improve estimates and the same for A0. Does one contribute more than the other for improving error rates?

**We have added this information (lines 357-358).**

Additional comments:
Line 86. Sensitivity saturates for some but not all taxa. See Haworth et al 2011.

**We have added the qualifier "in many species".**

Line 93. A Nearest living relative or equivalent approach also get around the issue of extinct taxa.

**This is true for the stomatal ratio method, but these CO2 estimates are not meant to be quantitative in the same manner as estimates from the "full calibration" methods or the gas-exchange methods (as noted in the previous paragraph).**

Line 156. Alternative approaches for fossils have been suggested such as estimating fossil A0 using scaling relationships between vein distance and assimilation rate however they are not discussed here (EG Montanez et al., 2016).

**We have added a citation to the Montanez paper**

Introduction – general comment. Critical published assessments of the Franks model are not cited (eg McElwain et al. 2016) yet they raise issues associated with parametrization of A0 and the insensitivity of CO2 estimates to variation in gamma star values which are both important discussion points in this manuscript in lines 454 -456 and 497-499.

**As per a later comment, we have added a citation to McElwain et al. 2016 regarding gamma star on line 472.**

**Our study does not focus on the parameterization of A0, and so the associated literature does not seem relevant to the Introduction. Our study focuses on temperature, photorespiration, canopy position, as well as a general and broad test of the method.**

Paragraph 201-217: A some information is missing here: chamber model/make, duration plants were grown in the chamber, light levels. What were measured vs set chamber conditions for temperature, light and CO2 (i.e. similar to how humidity is reported)

**Chamber make/model (lines 216-217) and duration of experiment (line 233) are given. We have added information about light intensity as well as the standard deviations for temperature and CO2 concentrations in lines 217-221.**

Lines 232: Stomatal density/stomatal measurements and leaf stable carbon isotopes were performed on the same leaves. Clarify how this was partitioned, e.g. was the leaf divided into 2 or was a whole punch used for carbon isotopes, etc.?

**We now clarify our methodology in lines 241-242. We used either a hole punch or razor to remove two adjacent sections of leaf tissue near the leaf centers, avoiding major veins.**

Lines 235: As Milligan et al is in review, I suggest adding more detail here on how δ13Ca of chamber CO2 was calculated. δ13Ca values of supplemented CO2 can be very negative and can vary between cylinders, unless the CO2 gas has a specific δ13Ca. What is the capacity of these cylinder, in L?

**This paper is now published. In short, a mixing line was established based on direct d13C measurements of lab air, chamber air, and cylinder CO2 (= pure CO2). We were fortunate that the d13C of the cylinder was close to the well-mixed atmosphere (the d13C in most cylinders we have used in other experiments is much more depleted). We used only the single cylinder for the duration of the experiment. The target CO2 concentration (500 ppm) was not much higher than the CO2 concentration inside the lab (~440 ppm), so we did not use much CO2.**

Figure 1: Does this need to be on a log scale? 1000 or 2000ppm are not very high values and the log scale visually skews data and error bars. A difference plot between measured and estimates plotted on a non-log scale would improve this figure.

**We prefer a log scale because it is easier to differentiate estimates at the low-end of the CO2 scale, and because the uncertainties scale in a logarithmic fashion.**

Line 351: Please provide supporting data for this statement in tabular form. What are the error rates of other proxies?

**This information was summarized by Franks et al. (2014), so we prefer not to repeat it here.**

Line 355: Might be helpful to report standard deviation of CO2 estimates, here and throughout the text.

**We now report the range that encompasses two-thirds of all estimates (lines 350-352). (Because the individual estimates are not normally distributed (tail at the high end), reporting a standard deviation can be misleading.)**

Line 411 to 413. Reporting of the difference between estimated and measured CO2 here is incomplete. Only means of all species investigated are provided rather than species-based diffeences or errors. For some species the error is substantial whereas other taxa show very small errors.

**As per an earlier comment, we now report the species-level differences on lines 425-427; no individual species-level test was significant (line 414).**

Line 454 to 456. This supports the findings of McElwain et al 2016 Paleo 3 but it is not cited. "This compensation point (Γ * in Eq. (2) is temperature, species and O2 dependent (Ethier and Livingston, 2004) but Franks et al. (2014) account only for the temperature dependency in the new paleo-CO2 proxy model. Allowing Γ * to vary in response to prevailing paleoatmospheric O2 concentration [O2] (Γ* = 1.78 × [O2]), which is known to have varied widely (10% to 30%) through the Phanerozoic (Bergman et al., 2004; Belcher and McElwain, 2008; Berner, 2009), would increase the precision of paleo-CO2 estimates but only fractionally."

**We have added a citation to McElwain et al. (2016 Palaeo3) (line 472).**

Lines 500 to 506: A number of papers have suggested methods of estimating A0 to improve the accuracy of CO2 estimates using the Franks model but they are not discussed. This section would provide a good opportunity to discuss the proposed ideas and solutions.

**This section deals with living leaves, where A could be measured directly. Measuring A wasn't part of our study design, unfortunately. In this section we are discussing possible reasons for noise in our mixing-model calculations. With regards to fossils, we are not recommending that our mixing model be used (line 527: "We note that our mixing-model strategy cannot be applied to fossils because…"), so the question of how to constrain A in fossils within the context of the mixing model is moot. Our take-home message for fossil applications is to avoid shade leaves (line 535), and we provide specific measurements that can be made on fossils to make this distinction, including vein density (lines 536-540).**

Section 3.4: Have any values for $\delta 13Ca$ been measured or are all calculated for this section? Is there any data set (from the literature or otherwise) this could be compared to? i.e. a dataset where known $\delta 13Ca$ is compared to itself when calculated as per the manuscript? This would strengthen this section. If $\delta 13Ca$ has only been calculated/inferred for this section without a comparison to measured $\delta 13Ca$ I think claims on the effect of $\delta 13Ca$ (or low canopy plants) on the model should be softened.

**We made no direct measurements of understory d13Ca (multiple measurements over a growing season, and at different daytime hours, would be needed to calculate a representative mean value). As the reviewer correctly notes, we instead are assuming a well-behaved two end-member mixing model. We have added a note of caution related to this on lines 508-511.**

Appendix: The authors used both known and general values for gc(op)/gc(max) and A0 to evaluate error rates but no measured values of either gc(op)/gc(max) or A0 are given in the appendix or text.

**The Appendix summarizes all new data presented in the study (with the key graphics being Figures 2, 5, and 7). For these data, we \*only\* used "default" values of gop/gmax and Ao; that is, we did not measure these inputs on our leaves. As noted in the Introduction, this was a purposeful strategy because we wanted to test the CO2 model in a manner that would be similar to how most (but not all) folks will be applying the model to fossils. A "worst-case" test, if you will.**

**In the Introduction, we do summarize some of the already-published data (Figure 1). For these estimates, either gop/gmax or A0 were measured, and in most cases both were measured (lines 143-146). These data are not in the Appendix because they are already published and are not central to our study.**

**As the reviewer noted, we did additionally "degrade" these estimates by re-doing them assuming default values for gop/gmax and A0. We did this so we could compare them more directly to our estimates (lines 355-357).**

[revised manuscript text omitted]